# Molecular mapping and functional validation of GLP-1R cholesterol binding sites in pancreatic beta cells

**Affiong Ika Oqua[1], Kin Chao[2], Liliane El Eid[1], Lisa Casteller[2], Billy P Baxter[3], Alba Miguéns-Gómez[4], Sebastian Barg[4], Ben Jones[3], Jorge Bernardino de la Serna[5]\*, Sarah L Rouse[2]\*, Alejandra Tomas[1]\***

[1]Section of Cell Biology and Functional Genomics, Division of Diabetes, Endocrinology and Metabolism, Department of Metabolism, Digestion and Reproduction, Imperial College London, London, United Kingdom; [2]Department of Life Sciences, Imperial College London, London, United Kingdom; [3]Section of Investigative Medicine, Division of Diabetes, Endocrinology and Metabolism, Department of Metabolism, Digestion and Reproduction, Imperial College London, London, United Kingdom; [4]Department of Medical Cell Biology, University of Uppsala, Uppsala, Sweden; [5]National Heart and Lung Institute, Imperial College London, London, United Kingdom

**\*For correspondence:**
j.bernardino-de-la-serna@ imperial.ac.uk (JBdlS); s.rouse@imperial.ac.uk (SLR); a.tomas-catala@imperial.ac.uk (AT)

**Competing interest:** The authors declare that no competing interests exist.

## eLife Assessment

The study presents a **valuable** finding on the role of cholesterol-binding sites on GLP-1 receptors although the clinical ramifications are unclear and not eminent at this point. Based on the detailed and persuasive responses provided by authors to the concerns raised by reviewers, the revised manuscript is improved substantially and is **convincing** enough in its scientific merit. The study is a good addition to the scientific community working on receptor biology and drug development for GLP-1 R.

**Abstract** G protein-coupled receptors (GPCRs) are integral membrane proteins which closely interact with their plasma membrane lipid microenvironment. Cholesterol is a lipid enriched at the plasma membrane with pivotal roles in the control of membrane fluidity and maintenance of membrane microarchitecture, directly impacting on GPCR stability, dynamics, and function. Cholesterol extraction from pancreatic beta cells has previously been shown to disrupt the internalisation, clustering, and cAMP responses of the glucagon-like peptide-1 receptor (GLP-1R), a class B1 GPCR with key roles in the control of blood glucose levels via the potentiation of insulin secretion in beta cells and weight reduction via the modulation of brain appetite control centres. Here, we unveil the detrimental effect of a high cholesterol diet on GLP-1R-dependent glucoregulation in vivo, and the improvement in GLP-1R function that a reduction in cholesterol synthesis using simvastatin exerts in pancreatic islets. We next identify and map sites of cholesterol high occupancy and residence time on active vs inactive GLP-1Rs using coarse-grained molecular dynamics (cgMD) simulations, followed by a screen of key residues selected from these sites and detailed analyses of the effects of mutating one of these, Val229, to alanine on GLP-1R-cholesterol interactions, plasma membrane behaviours, clustering, trafficking and signalling in INS-1 832/3 rat pancreatic beta cells and primary mouse islets, unveiling an improved insulin secretion profile for the V229A mutant receptor. This study (1) highlights the role of cholesterol in regulating GLP-1R responses in vivo; (2) provides a detailed map of GLP-1R - cholesterol binding sites in model membranes; (3) validates their functional relevance

in beta cells; and (4) highlights their potential as locations for the rational design of novel allosteric modulators with the capacity to fine-tune GLP-1R responses.

## Introduction

The GLP-1R, expressed in pancreatic beta cells amongst other tissues including brain, lung, stomach, and heart, is activated by the incretin peptide hormone GLP-1, secreted from enteroendocrine L-cells after food intake (*de Graaf et al., 2016*), to regulate postprandial glucose levels via the potentiation of glucose-dependent insulin secretion (*Boer and Holst, 2020*; *Gutierrez-Aguilar and Woods, 2011*; *Holst, 2007*). Activation of GLP-1R also promotes beta cell survival, inhibits gastric emptying, regulates food intake, and reduces appetite (*de Graaf et al., 2016*; *Drucker and Holst, 2023*), making it a key pharmacological target for various metabolic disorders including type 2 diabetes (T2D) and obesity (*El Eid et al., 2022*; *Miller et al., 2014*). Circulation of active GLP-1 (7–36) hormone is short-lived due to its rapid cleavage into inactive GLP-1 (9–36) by dipeptidyl peptidase-4 (DPP-4) *Boer and Holst, 2020*; DPP-4-resistant peptide analogues of GLP-1 have, therefore, been developed and successfully used clinically for the treatment of T2D and obesity, including exendin-4, liraglutide, and semaglutide, amongst others (*de Graaf et al., 2016*; *Boer and Holst, 2020*; *Drucker and Holst, 2023*). Despite their success, access to incretin peptide analogues is challenging for most individuals that require these therapies due to their high cost and complex manufacturing process leading to supply shortages, as well as the requirement for refrigeration and administration by injection, together with notable side effects including gastrointestinal disturbances, prompting further research into the development of a new wave of small molecules (*Camilleri and Acosta, 2024*; *Lu et al., 2023*), including those targeting the receptor as allosteric or ago-allosteric modulators (*Smelcerovic et al., 2019*; *Cong et al., 2021*).

The GLP-1R is part of the glucagon family of receptors, including the glucose-dependent insulinotropic polypeptide receptor (GIPR) (*Drucker and Holst, 2023*), the glucagon receptor (GCGR) (*Müller et al., 2017*), and the glucagon-like peptide-2 receptor (GLP-2R) (*Drucker and Yusta, 2014*), which are class B1/secretin-like GPCRs with a typical structure of seven transmembrane domains (7-TMD), a C-terminal tail, and a large N-terminal extracellular domain (ECD) important for peptide agonist binding (*Bortolato et al., 2014*). GPCRs transmit extracellular signals via coupling to heterotrimeric G proteins and interactions with β-arrestins, leading to activation of various intracellular signalling cascades (*Baccouch et al., 2022*); the GLP-1R in particular signals preferentially via $G\alpha_s$ coupling and adenylate cyclase activation, leading to cAMP accumulation and downstream signal transduction (*Marzook et al., 2021*). Their 7-TMD structure involves intimate interactions with plasma membrane lipids (*Baccouch et al., 2022*), known to be non-homogeneously distributed but rather organised in regions known as lipid nanodomains or lipid rafts, enriched in lipid species including sphingolipids and cholesterol, as well as in integral membrane proteins serving as scaffolds for signal transduction (*Sezgin et al., 2017*; *Bernardino de la Serna et al., 2016*), and for GPCR trafficking and sorting (*Baccouch et al., 2022*; *Kumar and Chattopadhyay, 2021*; *Sunshine and Iruela-Arispe, 2017*).

The plasma membrane-enriched lipid cholesterol has been identified as an allosteric modulator for certain GPCRs, due to its direct binding to *McGraw et al., 2022*; *Hanson et al., 2008* and modulation of GPCR activation, trafficking, and signalling patterns (*Sejdiu and Tieleman, 2020*; *Shrivastava et al., 2022*). A previous study from our laboratory established that the GLP-1R function in pancreatic beta cells is also modulated by cholesterol. Specifically, we showed that GLP-1R activation leads to receptor segregation into flotillin-positive lipid nanodomains and that disruption of the plasma membrane architecture by cholesterol extraction with methyl-β-cyclodextrin (MβCD) significantly reduces exendin-4-mediated receptor clustering, internalisation, and cAMP accumulation in beta cells (*Buenaventura et al., 2019*). Here, we focus on identifying the effect of more nuanced changes in cholesterol content on exendin-4-mediated GLP-1R function in primary islets and in vivo, followed by mapping of cholesterol binding sites for both apo-state and active receptor structures using coarse-grained molecular dynamics (cgMD) simulations tailored to identify lipid-receptor interactions, and functional validation of a prominent site for its effects on cholesterol binding, plasma membrane behaviours and exendin-4-mediated GLP-1R function in beta cells and primary islets. Our investigation identifies new potentially druggable locations in the receptor, expanding the existing repertoire

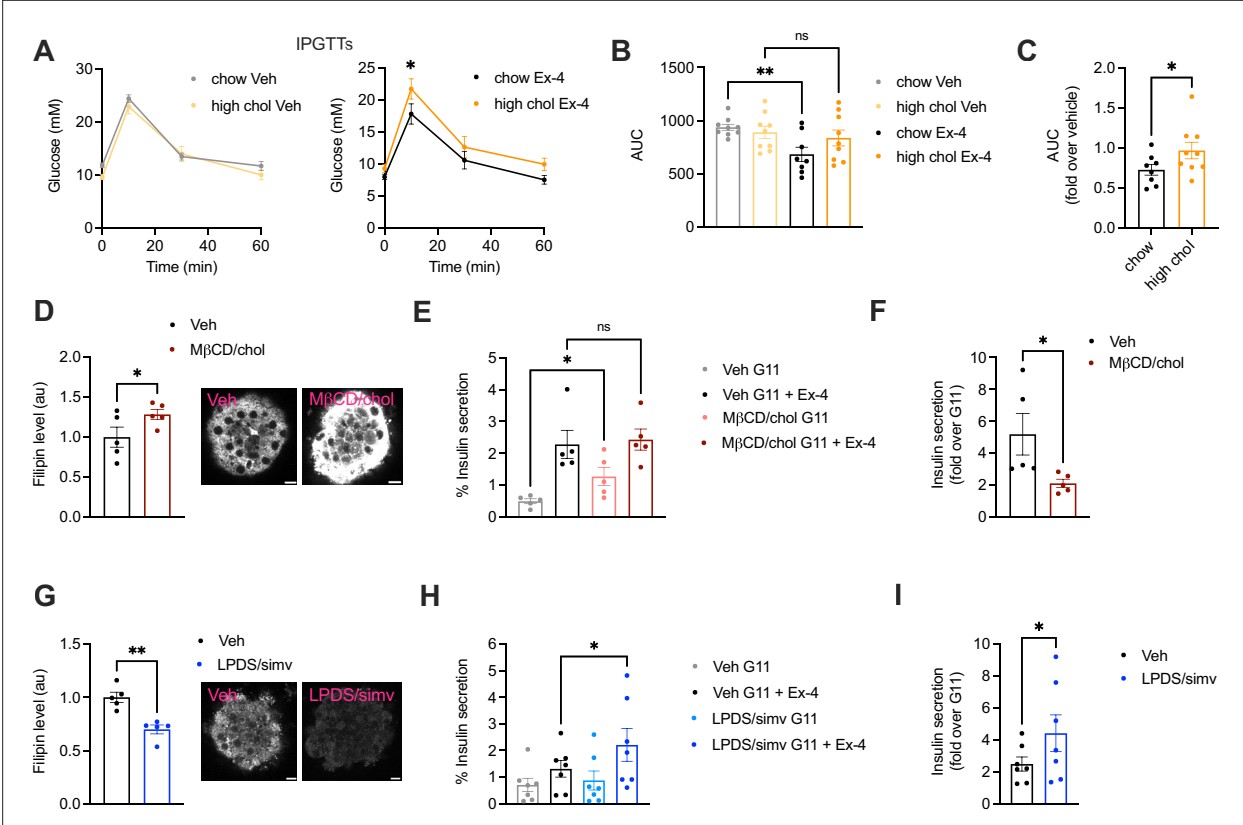

**Figure 1.** Effect of changes in cholesterol on glucagon-like peptide-1 receptor (GLP-1R) agonist responses in vivo and in primary islets.
(**A**) Intraperitoneal glucose tolerance tests (IPGTTs) 6 hr post-intraperitoneal administration of vehicle (Veh) or 1 nmol/kg exendin-4 (Ex-4) in mice fed a chow vs 2% cholesterol (high chol) diet for 5 wk; n=8–9 female mice per diet. (**B**) Area under the curve (AUC) for glucose curves from (**A**). (**C**) Ex-4 over Veh glucose levels in chow vs 2% cholesterol diet fed mice. (**D**) Average intensity of filipin staining (to label cholesterol) in mouse islets preincubated with Veh or methyl-β-cyclodextrin (MβCD) loaded with 20 mM cholesterol (MβCD/chol) for 1 hr; n=5 islet preps from separate mice; representative islet images also shown; size bars, 10 μm. (**E**) Percentage of insulin secretion from mouse islets preincubated with Veh or MβCD/chol before stimulation with 11 mM glucose (**G11**) +/-100 nM Ex-4; n=5. (**F**) Ex-4-induced insulin secretion (fold over G11) in mouse islets from (**E**). (**G**) Average filipin staining in mouse islets preincubated with Veh or lipoprotein-deficient serum (LPDS) media supplemented with 10 μM simvastatin (LPDS/simv) overnight; n=5 islet preps from separate mice; representative islet images also shown; size bars, 10 μm. (**H**) Percentage of insulin secretion from mouse islets preincubated with Veh or LPDS/simv before stimulation with G11 +/-100 nM Ex-4; n=7. (**I**) Ex-4-induced insulin secretion (fold over G11) in mouse islets from (**H**). Data is mean +/- SEM; ns, non-significant, *p<0.05, **p<0.01 by paired t-test or one-way ANOVA with Sidak's multiple comparison test.

The online version of this article includes the following figure supplement(s) for figure 1:

**Figure supplement 1.** Effect of changes in cholesterol on glucagon-like peptide-1 receptor (GLP-1R) agonist responses in primary islets - contd.

of functionally relevant sites for the rational design of novel GLP-1R allosteric modulators based on the modulation of receptor-cholesterol interactions.

## Results

### Changes in cholesterol content modulate exendin-4-induced GLP-1R behaviours in primary islets and in vivo

To determine the effect of increased cholesterol levels on GLP-1R function, we fed mice a standard chow diet supplemented or not with cholesterol for 5 wk. We then performed intraperitoneal glucose tolerance tests (IPGTTs) post-administration of vehicle or 1 nmol/kg exendin-4 to determine the glucose lowering effect of pharmacologically targeting the receptor under these conditions. Mice on either diet had similar glucose responses under vehicle conditions; however, those on high cholesterol diet presented worse glucose lowering responses to exendin-4 vs chow-fed mice (*Figure 1A and B*), an effect that was maintained when normalised to the corresponding vehicle control (*Figure 1C*). Islets isolated from high cholesterol-fed mice displayed a near-significant increase in cholesterol

(*Figure 1—figure supplement 1A*) accompanied by decreased exendin-4-induced cAMP responses ex vivo (*Figure 1—figure supplement 1B*), highlighting the negative impact of even mild increases in cholesterol on GLP-1R signalling.

To more directly evaluate the effect of increased islet cholesterol content on GLP-1R function, islets isolated from wild-type (WT) mice were loaded ex vivo with MβCD saturated with cholesterol, which led to a significant increase in cholesterol content (*Figure 1D*). Under these conditions, we measured a decrease in insulin secretion in response to exendin-4 *vs* vehicle conditions, primarily due to elevated basal secretion levels at 11 mM glucose (*Figure 1E and F*), confirming that increased cholesterol levels reduce the GLP-1R glucoregulatory potential by decreasing its capacity to potentiate islet insulin secretion. This lack of proportional secretory increase following exendin-4 stimulation of cholesterol-loaded islets was not related to an overall saturation of the secretory response, as the percentage of insulin secretion was further increased in both vehicle and cholesterol-loaded islets by stimulation with a secretagogue cocktail (*Figure 1—figure supplement 1C*). These effects were accompanied by a decreased capacity for exendin-4 to reduce GLP-1R plasma membrane diffusion (*Figure 1—figure supplement 1D, E*), a behaviour observed here and previously (*Manchanda et al., 2023*; *Pickford et al., 2020*) by Raster Image Correlation Spectroscopy (RICS) in rat insulinoma INS-1 832/3 cells without endogenous GLP-1R [INS-1 832/3 GLP-1R KO cells (*Naylor et al., 2016*)] stably expressing SNAP/FLAG-tagged human GLP-1R (SNAP/FLAG-hGLP-1R).

We next performed the reverse experiment by decreasing islet cholesterol levels with simvastatin, a long-established hydroxy-methylglutaryl coenzyme A reductase (HMGCR) inhibitor (*Pedersen and Tobert, 2004*). Filipin staining confirmed decreased islet cholesterol content (*Figure 1G*), which correlated with increased exendin-4-induced cAMP responses (*Figure 1—figure supplement 1F*), and exendin-4-induced insulin secretion (*Figure 1H, I*) in simvastatin-exposed islets. Contrary to cholesterol loading, simvastatin triggered decreased GLP-1R plasma membrane diffusion under vehicle conditions, measured by RICS as above (*Figure 1—figure supplement 1G, H*), highlighting the importance of cholesterol in determining GLP-1R movement at the plasma membrane. Taken together, these results demonstrate that changes in cholesterol levels have specific effects on beta cell GLP-1R behaviours, indicating the potential existence of direct interactions between the receptor and this plasma membrane-enriched lipid.

## Coarse-grained molecular dynamics simulations reveal GLP-1R cholesterol binding sites and state-dependent cholesterol binding modes

To gain insights into the precise location of cholesterol interactions with the GLP-1R, cgMD simulations were employed to study both active (corresponding to agonist bound) and inactive (apo-) states of GLP-1R in a model mammalian plasma membrane using the latest Martini 3 forcefield and cholesterol parameters (*Figure 2A*). The PyLipID package (*Song et al., 2022*) was employed to characterise cholesterol interactions for each receptor state. Two key parameters were measured: occupancy, defined as the percentage of frames in the trajectory where any cholesterol molecule is within the cutoff distance from the receptor, and interaction residence time, defined as the time that a cholesterol molecule remains in the binding site, providing information about the strength of the interaction (full data included in *Supplementary file 1* and *Supplementary file 2*). Occupancy values for each receptor residue in both states were calculated and displayed as heatmaps, with the top 10 residues per state indicated in *Figure 2B*. The receptor active and inactive states exhibited similar top occupancy residues, with hydrophobic residues such as Phe, Leu, Ile, and Val enriched in both states. A generally similar profile was observed for the top 30 highest occupancy residues in active *vs* inactive states, with some subtle differences noted in the receptor transmembrane domains TM2 and TM6 (*Figure 2C*).

The top three cholesterol binding sites, displaying the highest cholesterol residence times for both states, were calculated using the PyLipID package (*Figure 2D and E*). Interestingly, notable differences were observed in these sites between the two receptor states: for the active state, site I (purple) was located in the upper TM1 and TM7 region, while in the inactive state, site I (cyan) was found in the lower TM5 and TM6 region, an area which undergoes key structural changes upon G protein engagement and GLP-1R activation (*Zhang et al., 2017*) and might, therefore, represent a state-specific cholesterol binding site. Site II (orange) was similar for both states but presented subtle

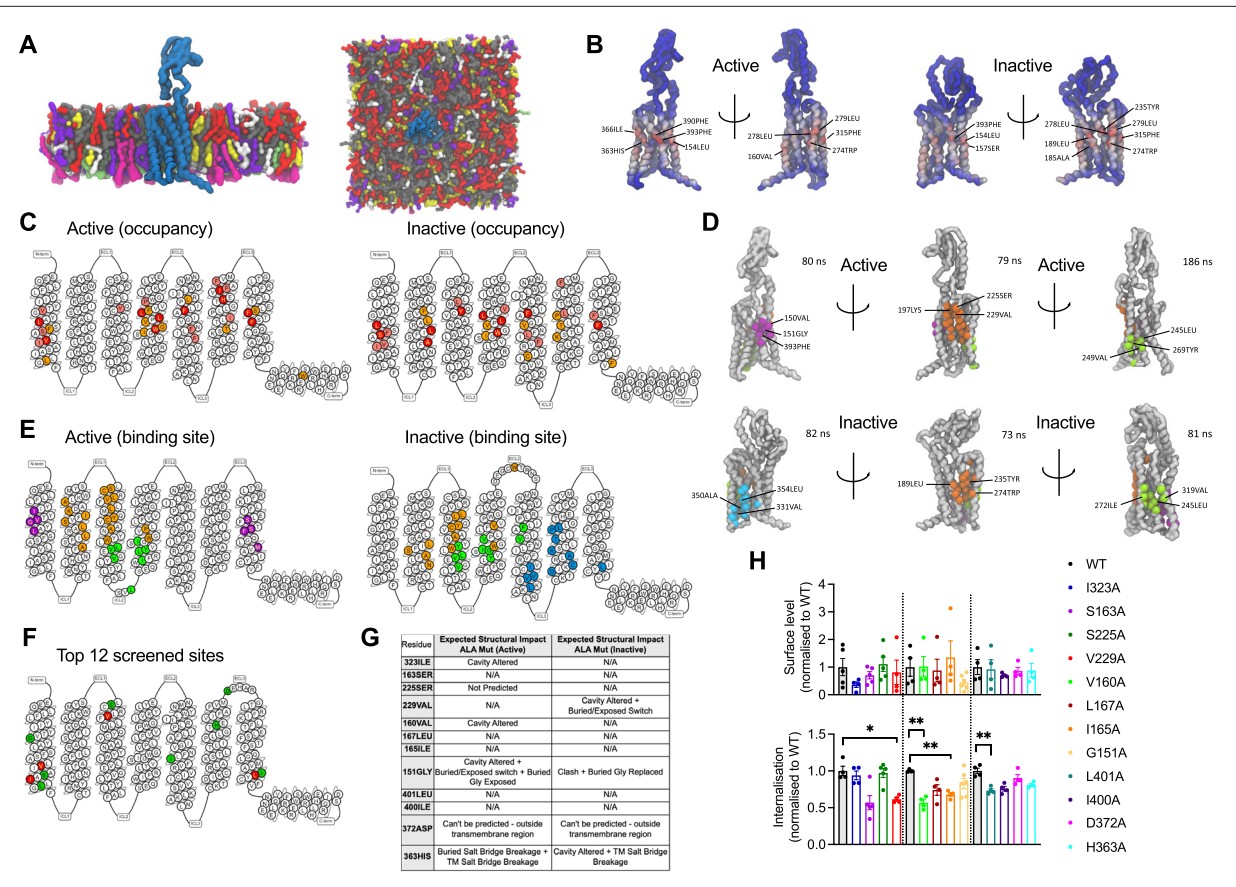

**Figure 2.** cgMD simulations of glucagon-like peptide-1 receptor (GLP-1R) – cholesterol binding sites in model membranes. (**A**) Overview of the simulation setup - GLP-1R is embedded in a model mammalian plasma membrane with the following composition: POPC (30%), DOPC (30%), POPE (8%), DOPE (7%), and cholesterol (25%) in the upper leaflet, and POPC (5%), DOPC (5%), POPE (20%), DOPE (20%), POPS (8%), DOPS (7%), PIP$_2$ (10%), and cholesterol (10%) in the lower leaflet. (**B**) Average cholesterol occupancy profile in active (left) and inactive (right) GLP-1R states shown as a heatmap (red – highest occupancy; blue – lowest occupancy), with the top 10 highest occupancy residues per state labelled. (**C**) Snake plot showing the top 30 highest cholesterol occupancy residues in active and inactive states, with colours indicating occupancy levels (top 10 – red; top 20 – pink; top 30 – orange). (**D**) Top three cholesterol binding sites in GLP-1R active (top) vs inactive (bottom) states, calculated using PyLipID. Binding sites are colour-coded as follows: site I - purple in active and cyan in inactive state; site II - orange; site III - green, with the top 3 residues with the highest residence time in each site labelled and average residence time indicated for each site. (**E**) GLP-1R snake plot indicating residues from top three cholesterol binding sites in both states using the same colour scheme as in (**D**). (**F**) GLP-1R snake plot indicating the 12 residues selected for screening, with the 4 residues showing a significant reduction in GLP-1R internalisation when mutated to alanine (see H) coloured in red, and the remaining residues in green. (**G**) Table showing the predicted structural impact of the 12 selected residues after site-directed mutagenesis to alanine in active vs inactive GLP-1R using Missense3D-TM (*Hanna et al., 2024*). (**H**) Surface expression and exendin-4 (100 nM, 10 min) mediated internalisation screen of the 12 selected residues from GLP-1R-cholesterol binding sites mutated to alanine, transiently transfected in INS-1 832/3 GLP-1R KO cells; n=4–5. Data is mean +/- SEM, *p<0.05, **p<0.01 by one way ANOVA with Dunnett's multiple comparison test vs corresponding wild-type (WT) SNAP/FLAG-hGLP-1R.

differences, with an additional cholesterol pocket involving the Trp in extracellular loop 2 (ECL2) being more pronounced in the inactive vs the active state. Site III (green) showed some degree of variation between the two states; in the inactive state, two TM5 residues (Val319 and Phe315) formed an additional cholesterol pocket with TM4, presenting the highest average residence time (186 nanoseconds) across all sites, while, in the active state, the cholesterol pocket was located closer to the bottom of TM3 and TM4, involving Leu260 in ECL2.

To evaluate the functional relevance of the GLP-1R cholesterol binding sites identified during the cgMD simulations, 12 residues were selected within or at the periphery of these sites and mutated to alanine by site-directed mutagenesis (*Figure 2F*). Alanine mutations for four of the selected residues (Ile323, Val160, Gly151, and His363) were predicted to cause changes in the GLP-1R structure in the active state by altering the cavity or causing changes in exposed residues and breakages in salt bridges between residues, while a further three alanine mutated residues (Val229, Gly151, and His363)

were predicted to cause changes in the inactive GLP-1R structure (*Figure 2G*). SNAP/FLAG-tagged human GLP-1Rs (SNAP/FLAG-hGLP-1Rs) with single alanine substitutions for each selected residue were successfully expressed in INS-1 832/3 GLP-1R KO cells with no significant differences detected in cell surface expression levels *vs* WT GLP-1R (*Figure 2H*, top graph). Mutant receptors were then screened for their capacity to undergo exendin-4-induced internalisation, as this GLP-1R property is severely inhibited following cholesterol extraction with MβCD (*Buenaventura et al., 2019*). Four alanine mutated residues, V229A, V160A, I165A, and L401A, caused a significant decrease in GLP-1R internalisation *vs* WT receptor, with GLP-1R V229A and V160A causing a ~40%, and I165A and L401A a ~30% decrease in internalisation (*Figure 2H*, bottom graph). GLP-1R V229A was selected for further detailed analysis of the effects of modifying a GLP-1R-cholesterol binding site residue on GLP-1R beta cell behaviours.

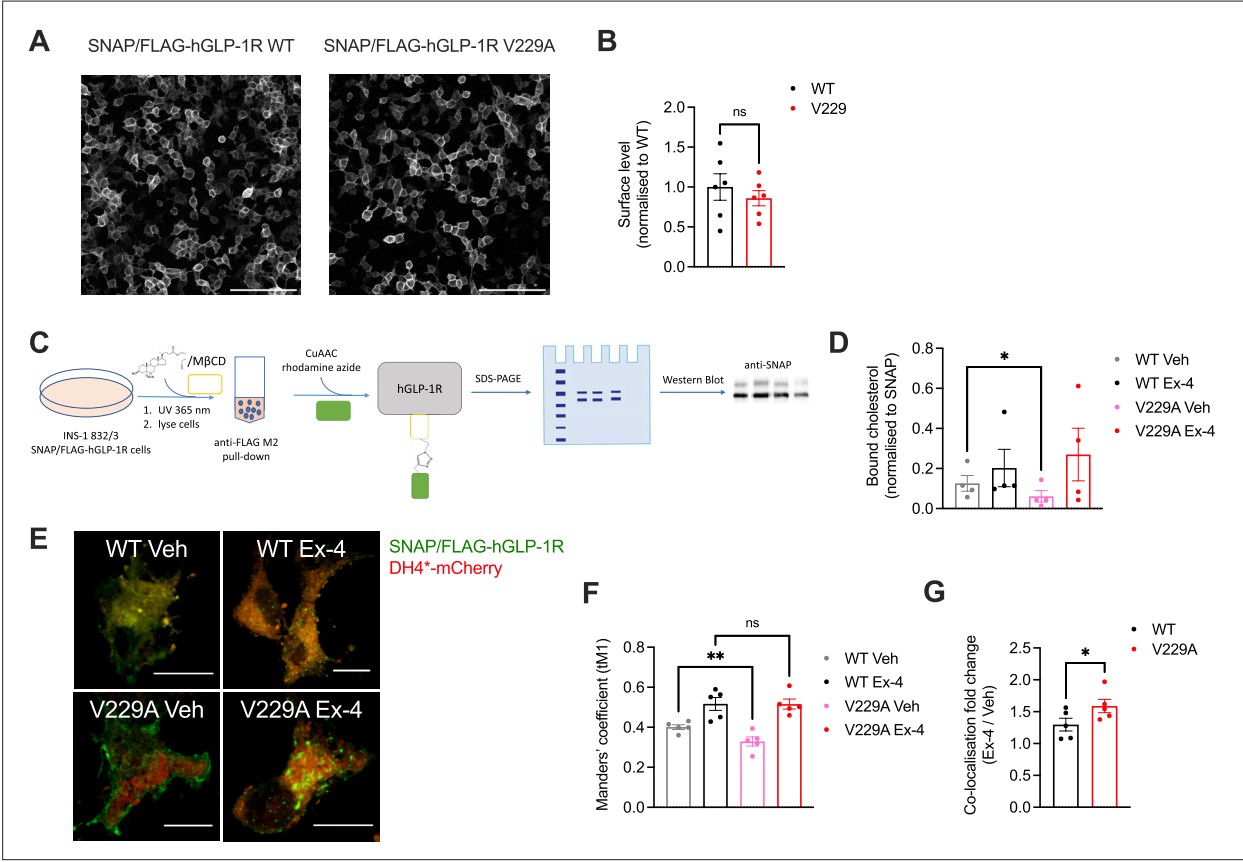

**Figure 3.** Glucagon-like peptide-1 receptor (GLP-1R) wild-type (WT) *vs* V229A cholesterol binding propensity. (**A**) Representative images of INS-1 832/3 SNAP/FLAG-hGLP-1R WT or V229A sublines labelled with SNAP-Surface Alexa Fluor 647; size bars, 100 µm. (**B**) Surface expression of SNAP/FLAG-hGLP-1R WT *vs* V229A; n=6. (**C**) Schematic diagram of the GLP-1R PhotoClick cholesterol binding assay. (**D**) SNAP/FLAG-hGLP-1R-bound cholesterol normalised to receptor levels in INS-1 832/3 SNAP/FLAG-hGLP-1R WT or V229A treated with vehicle (Veh) or 100 nM exendin-4 (Ex-4) for 2 min; n=4. (**E**) Representative images of INS-1 832/3 SNAP/FLAG-hGLP-1R WT *vs* V229A cells labelled with SNAP-Surface 488 (green) and stimulated with Veh *vs* Ex-4 for 2 min prior to fixation and labelling with D4H*-mCherry (red); size bars, 5 µm. (**F**) Quantification of co-localisation (Mander's tM1) between SNAP/FLAG-hGLP-1R WT or V229A and D4H*-mCherry in cells from (**E**); n=5. (**G**) Ex-4 over Veh co-localisation fold change for WT *vs* V229A SNAP/FLAG-hGLP-1R; n=5. Data is mean +/- SEM, ns, non-significant, *p<0.05, **p<0.01 by paired t-test or one-way ANOVA with Sidak's multiple comparison test.

The online version of this article includes the following source data and figure supplement(s) for figure 3:

**Figure supplement 1.** Glucagon-like peptide-1 receptor (GLP-1R) wild-type (WT) *vs* V229A cholesterol binding, and conformational shift - contd.

**Figure supplement 1—source data 1.** PDF file containing original western blots for *Figure 3—figure supplement 1A*, indicating the relevant bands and treatments.

**Figure supplement 1—source data 2.** Original files for western blot analysis displayed in *Figure 3—figure supplement 1A*.

## GLP-1R V229A substitution affects GLP-1R-cholesterol interactions and plasma membrane behaviours in pancreatic beta cells

We next generated INS-1 832/3 GLP-1R KO sublines stably expressing SNAP/FLAG-hGLP-1R WT or V229A. As expected, no difference in surface expression was detected between WT and V229A receptors, although we observed a non-significant tendency for reduced surface levels with the V229A mutant (*Figure 3A and B*). To determine the effect of the V229A substitution on the overall GLP-1R binding to cholesterol in active *vs* inactive conditions, cells from both sublines were labelled with PhotoClick cholesterol and exposed to UV light prior to SNAP/FLAG-hGLP-1R purification by anti-FLAG immunoprecipitation and click chemistry to fluorescently label covalently bound PhotoClick cholesterol (*Figure 3C*). Resulting samples were separated by SDS-PAGE and gels imaged under fluorescence followed by anti-SNAP Western blotting (*Figure 3—figure supplement 1A*). Quantification of GLP-1R-bound cholesterol in both WT and V229A receptors following vehicle *vs* acute (2 min) exendin-4 stimulation showed a significant decrease in cholesterol binding to GLP-1R V229A in vehicle conditions without modifying cholesterol binding after agonist stimulation compared to GLP-1R WT (*Figure 3D*). Co-localisation of SNAP/FLAG-hGLP-1R with cholesterol in the same cells using the fluorescent D4H*-mCherry probe to label plasma membrane cholesterol at the cytosolic leaflet (*Lim et al., 2019*) corroborated our previous results as we measured a significant decrease in

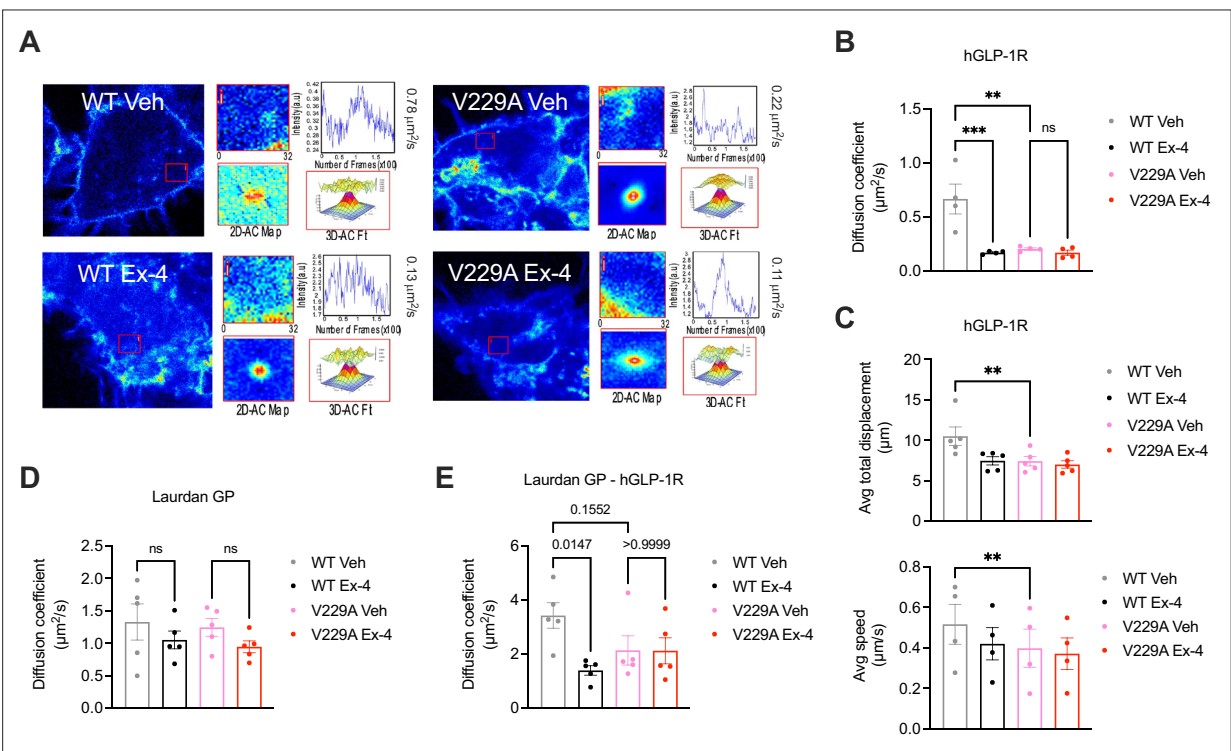

**Figure 4.** Glucagon-like peptide-1 receptor (GLP-1R) wild-type (WT) *vs* V229A movement at the plasma membrane. (**A**) Representative images from GLP-1R WT *vs* V229A raster image correlation spectroscopy (RICS) analysis of plasma membrane lateral diffusion in INS-1 832/3 SNAP/FLAG-hGLP-1R WT or V229A cells labelled with SNAP-Surface Alexa Fluor 647 before stimulation with vehicle (Veh) or 100 nM exendin-4 (Ex-4). (**B**) Average RICS diffusion coefficients from GLP-1R WT *vs* V229A from (**A**); n=4. (**C**) TIRF-SPT analysis of average total displacement (top) and speed (bottom) of GLP-1R WT *vs* V229A under Veh or Ex-4-stimulated conditions in INS-1 832/3 GLP-1R KO cells expressing hGLP-1R-mEGFP WT *vs* V229A; n=4 for total displacement and n=5 for speed. (**D**) Average RICS diffusion coefficients of plasma membrane lipid-ordered nanodomains (measured as changes in membrane fluidity/lipid packing assessed by Laurdan GP values) in INS-1 832/3 SNAP/FLAG-hGLP-1R WT *vs* V229A cells under Veh or Ex-4-stimulated conditions; n=5. (**E**) Average diffusion coefficient from RICCS analysis of SNAP-Surface Alexa Fluor 647-labelled SNAP/FLAG-hGLP-1R WT *vs* V229A with lipid-ordered nanodomains assessed as in (**D**) under Veh or Ex-4-stimulated conditions in INS-1 832/3 SNAP/FLAG-hGLP-1R WT *vs* V229A cells; n=5. Data is mean +/- SEM; ns, non-significant, **p<0.01, ***p<0.001 by one-way ANOVA with Sidak's multiple comparison test.

The online version of this article includes the following figure supplement(s) for figure 4:

**Figure supplement 1.** glucagon-like peptide-1 receptor (GLP-1R) wild-type (WT) *vs* V229A lipid cross-correlation - contd.

co-localisation between GLP-1R V229A and cholesterol under vehicle conditions compared to WT GLP-1R (*Figure 3E–G*).

While we could not measure any changes in receptor conformational shift following exendin-4 binding between GLP-1R WT and V229A (*Figure 3—figure supplement 1B*, C), or differences in binding affinity to the fluorescent exendin-4 derivative exendin-asp3-TMR (*Figure 3—figure supplement 1D*), quantification of GLP-1R plasma membrane diffusion using RICS in INS-1 832/3 SNAP/FLAG-hGLP-1R WT *vs* V229A cells revealed a pronounced reduction in GLP-1R V229A diffusion compared to WT GLP-1R under vehicle conditions (*Figure 4A and B*), to a similar extent to the reduction caused by exendin-4 binding to GLP-1R WT, recapitulating the effect triggered by reduced cholesterol levels following simvastatin exposure (*Figure 1—figure supplement 1G, H*). Single particle tracking (SPT) of human GLP-1R fused to monoclonal EGFP (hGLP-1R-mEGFP) WT *vs* V229A imaged by total internal reflection fluorescence (TIRF) microscopy also showed reduced total displacement and speed (displacement over time) of V229A *vs* WT receptors under vehicle conditions, with these parameters also decreased following WT GLP-1R stimulation with exendin-4 (*Figure 4C*). Finally, exendin-4-induced activation of either WT or V229A GLP-1R tended to reduce the diffusion of lipid-ordered nanodomains at the plasma membrane, measured via RICS analysis of generalised polarisation (GP) values obtained from the environmentally sensitive lipid probe Laurdan (*Brewer et al., 2010*; *Figure 4D*), with no differences in Laurdan GP lateral diffusion in cells expressing WT *vs* V229A receptors. However, RICCS cross-correlation analysis of lipid-ordered nanodomain co-diffusion with GLP-1R WT *vs* V229A showed a reduction in the movement of GLP-1R-lipid nanodomain complexes for WT receptors after exendin-4 stimulation, with a tendency for V229A receptors to exhibit the same behaviour already under vehicle conditions, resulting in no significant change for GLP-1R V229A-lipid

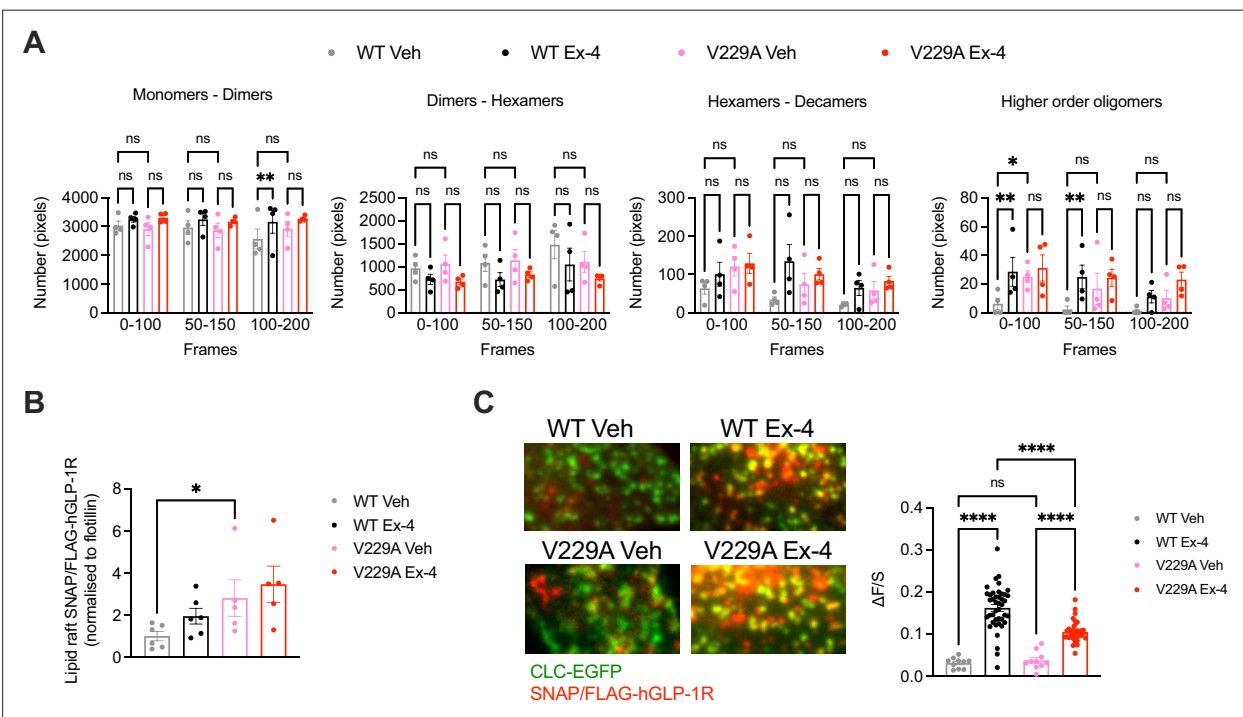

**Figure 5.** Glucagon-like peptide-1 receptor (GLP-1R) wild-type (WT) *vs* V229A oligomerisation and clathrin-coated pit (CCP) recruitment. (**A**) N&B estimation of average number of pixels for the different oligomerisation states of the GLP-1R, either as monomers-dimers, dimers-hexamers, hexamers-decamers, or higher order oligomers, calculated at different time frames after stimulation with either vehicle (Veh) or 100 nM exendin-4 (Ex-4) from INS-1 832/3 SNAP/FLAG-hGLP-1R WT or V229A cells; n=4. (**B**) GLP-1R WT *vs* V229A levels at lipid raft fractions purified from INS-1 832/3 SNAP/FLAG-hGLP-1R WT *vs* V229A cells under Veh or Ex-4-stimulated conditions. Results represent SNAP levels assessed by Western blotting normalised to flotillin as a marker of lipid raft enrichment; n=5–6. (**C**) Left: representative TIRF images of INS-1 832/3 SNAP/FLAG-hGLP-1R WT or V229A cells co-expressing clathrin light chain-GFP (CLC-GFP) labelled with SNAP-Surface Alexa Fluor 647 prior to Veh or 100 nM Ex-4 stimulation; right: quantification of association (ΔF/S, see Methods) of SNAP/FLAG-hGLP1-R WT or V229A with clathrin puncta; n=29 and 127 cells for WT Veh *vs* Ex-4, and n=31 and 106 cells for V229A Veh *vs* Ex-4, respectively; each data point represents mean of 3 cells, data collated from three separate experiments. Data is mean +/-SEM; ns, non-significant, *p<0.05, **p<0.01, ****p<0.0001 by unpaired t-test, one- or two-way ANOVA with Sidak's multiple comparison test.

nanodomain complex diffusion between vehicle and exendin-4-stimulated conditions (*Figure 4E*, *Figure 4—figure supplement 1*).

Number and Brightness (N&B) analysis of the RICCS cross-correlation data was then used to determine the effect of the V229A substitution on receptor oligomerisation before and after exendin-4 stimulation (*Figure 5A*). Both WT and V229A receptors existed predominantly as monomers or dimers at a proportion of ~60 to 80% of total receptors, a percentage that was reduced for the WT receptor at later acquisition times under vehicle conditions, but with otherwise few changes following exendin-4 stimulation. Around 30% of inactive WT or V229A receptors existed as dimers to hexamers or hexamers to decamers, with a tendency for the dimer to hexamer population to reduce and the hexamer to decamer population to increase after exendin-4 stimulation, and for GLP-1R V229A to have a higher proportion of hexamers to decamers under vehicle conditions *vs* WT receptors. A smaller proportion of receptors existed as higher order oligomers, with a significant increase measured for the WT GLP-1R after stimulation with exendin-4. For the GLP-1R V229A, however, we observed a higher proportion of receptors pre-existing as higher order oligomers under vehicle conditions, with no further increases after exendin-4 stimulation, indicating that the mutant receptor has a higher clustering propensity in the inactive state. Consistent with this finding, GLP-1R V229A also presented with significantly increased levels of receptors segregated to lipid rafts (*Lorizate et al., 2021*) prior to exendin-4 stimulation *vs* GLP-1R WT (*Figure 5B*).

To determine if the V229A substitution affected receptor recruitment to clathrin-coated pits (CCPs) after agonist stimulation, TIRF images were acquired of both unstimulated and exendin-4-stimulated INS-1 832/3 SNAP/FLAG-hGLP-1R WT *vs* V229A cells (*Figure 5C*). Before agonist addition, the SNAP/FLAG-hGLP-1R signal was distributed across the plasma membrane and only weakly associated with clathrin-positive regions (ΔF/S=0.03±0.003 and 0.04±0.006 for WT and V229A, respectively), with no differences between WT and V229A receptors. However, after exendin-4 stimulation, the receptor association with clathrin-positive regions increased, an effect that correlates with the known rapid GLP-1R internalisation triggered by exendin-4 (*Buenaventura et al., 2019*), with this increase, however, significantly less pronounced for the GLP-1R V229A compared to WT (ΔF/S=0.16±0.006 and 0.10±0.004 for WT *vs* V229A, respectively), in agreement with the reduced internalisation propensity displayed by this mutant receptor (*Figure 2H*).

## GLP-1R V229A substitution affects receptor trafficking, activation, and signalling in pancreatic beta cells

To determine the effects of the GLP-1R V229A substitution on GLP-1R trafficking, we assessed GLP-1R internalisation, recycling, and degradation parameters in INS-1 832/3 SNAP/FLAG-hGLP-1R WT *vs* V229A cells. GLP-1R V229A internalised more slowly than WT receptor after exendin-4 stimulation (*Figure 6A–C*), with only ~10% of V229A internalised after 5 min stimulation *vs* ~30% for WT. V229A internalisation eventually recovered after 30 min of exendin-4 exposure, reaching ~50% for both receptor types. In contrast, plasma membrane recycling of internalised GLP-1R V229A was increased *vs* WT at both 1 hr and 3 hr recycling periods (*Figure 6D and E*), while no changes in exendin-4-induced GLP-1R degradation were detected between GLP-1R WT and V229A (*Figure 6F and G*).

Next, we performed NanoBiT complementation assays to determine the effect of the GLP-1R V229A substitution on receptor coupling to G proteins and β-arrestins, assessing mini-G$_s$, mini-G$_q$, and β-arrestin 2 recruitment dose response curves to WT *vs* V229A receptors after stimulation with a range of exendin-4 doses. Results showed a significant increase in both the potency and efficacy of mini-G$_s$ recruitment to V229A *vs* WT GLP-1R (*Figure 7A*). Conversely, mini-G$_q$ recruitment was not changed between WT and V229A GLP-1R (*Figure 7B*). Additionally, GLP-1R V229A caused a small but consistent increase in β-arrestin 2 recruitment compared to GLP-1R WT (*Figure 7C*). Bias calculations between mini-G$_s$ and β-arrestin 2 recruitment showed that GLP-1R V229A is Gα$_s$-biased *vs* WT GLP-1R (*Figure 7D*). We next performed bystander complementation assays to measure the recruitment of nanobody 37 (Nb37)-SmBiT, which recognises and binds to active Gα$_s$, to the plasma membrane (labelled with KRAS CAAX motif-LgBiT), or to endosomes (labelled with Endofin FYVE domain-LgBiT) in response to GLP-1R stimulation with exendin-4 (see *Figure 7—figure supplement 1* for a schematic of the assay). GLP-1R V229A displayed increased activity at the plasma membrane (*Figure 7E*), with a tendency for increased activity also from endosomes without reaching statistical significance (*Figure 7F*).

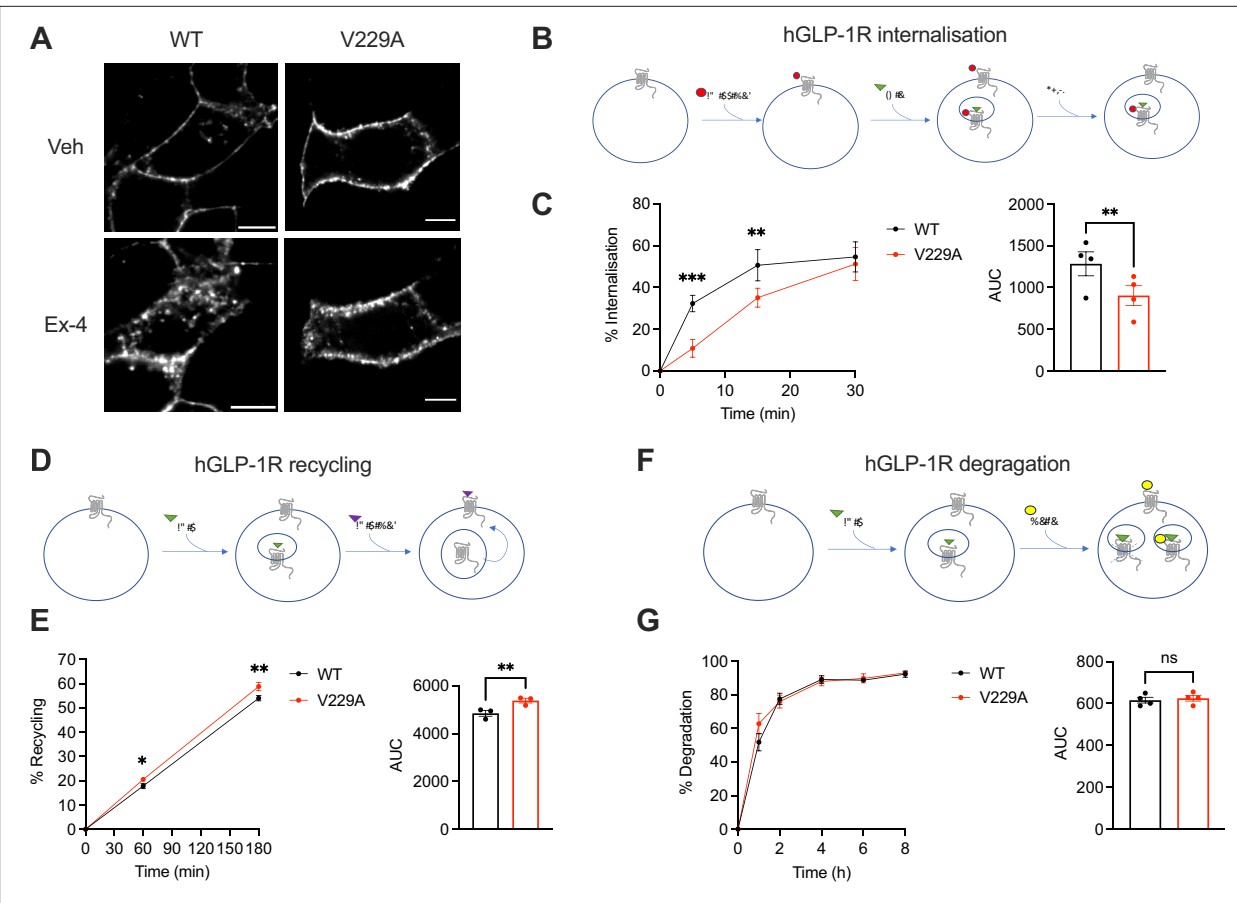

**Figure 6.** Glucagon-like peptide-1 receptor (GLP-1R) wild-type (WT) *vs* V229A trafficking profiles. (**A**) Representative images of INS-1 832/3 SNAP/FLAG-hGLP-1R WT or V229A cells labelled with SNAP-Surface Alexa Fluor 647 probe under vehicle (Veh) conditions or following stimulation with 100 nM exendin-4 (Ex-4) for 10 min; size bars, 5 µm. (**B**) Schematic diagram of agonist-mediated SNAP/FLAG-hGLP-1R internalisation assay. (**C**) Percentage of internalised SNAP/FLAG-hGLP-1R WT *vs* V229A at the indicated time points after stimulation with 100 nM Ex-4; corresponding area under the curve (AUC) also shown; n=4. (**D**) Schematic diagram of SNAP/FLAG-hGLP-1R plasma membrane recycling assay. (**E**) Percentage of recycled SNAP/FLAG-hGLP-1R WT *vs* V229A at the indicated time points after stimulation with 100 nM Ex-4; corresponding AUC also shown; n=3. (**F**) Schematic diagram of agonist-mediated SNAP/FLAG-hGLP-1R degradation assay. (**G**) Percentage of SNAP/FLAG-hGLP-1R WT *vs* V229A degradation at the indicated time points after stimulation with 100 nM Ex-4; corresponding AUC also shown; n=4. Data is mean +/- SEM; ns, non-significant, *p<0.05, **p<0.01, ***p<0.001 by paired t-test or two-way ANOVA with Sidak's multiple comparison test.

To determine differences between GLP-1R WT and V229A beta cell downstream signalling, cAMP and insulin secretion responses to exendin-4 were investigated in INS-1 832/3 SNAP/FLAG-hGLP-1R WT and V229A. GLP-1R V229A displayed a tendency for increased exendin-4-induced cAMP AUC, with kinetic responses that appeared delayed but reached significantly higher maximal cAMP peak levels (*Figure 8A and B*). Compared to WT receptor, GLP-1R V229A caused a significant decrease in basal insulin secretion under vehicle (11 mM glucose alone) conditions (*Figure 8C*), which resulted in significantly increased insulin secretion fold-responses to exendin-4 (*Figure 8D*). We then determined the effect of GLP-1R V229A in primary mouse islets. To this end, SNAP/FLAG-hGLP-1R WT or V229A-expressing adenoviruses were transduced into islets extracted from mice lacking endogenous GLP-1R expression (GLP-1R KO, generated *in house*). Similarly to the stable INS-1 832/3 sublines, there was no difference in surface expression of GLP-1R WT *vs* V229A in these islets, although again the V229A mutant receptor displayed a tendency for reduced cell surface levels (*Figure 8E*, top row and F). GLP-1R V229A also caused a significant decrease in receptor internalisation *vs* WT GLP-1R following exendin-4 stimulation (*Figure 8E*, bottom row and G). Expression of GLP-1R V229A was again associated with significantly improved insulin secretion fold-increases to exendin-4, but in this case, no changes were detected under 11 mM glucose alone conditions, with the improved fold being a consequence of enhanced insulin secretion *vs* WT GLP-1R during the exendin-4 stimulation period

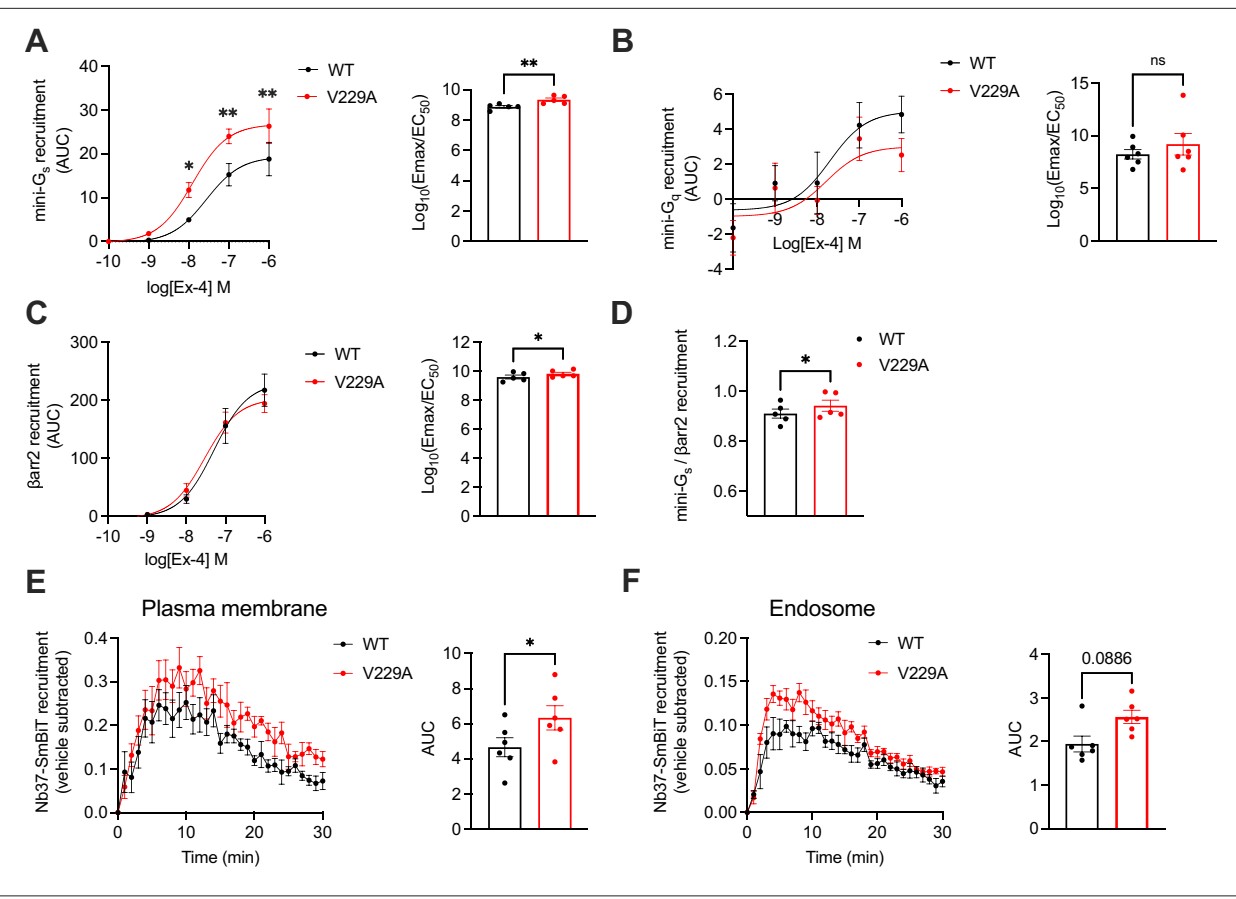

**Figure 7.** Signalling profiles of glucagon-like peptide-1 receptor (GLP-1R) wild-type (WT) *vs* V229A. (**A**) Mini-$G_s$ recruitment dose response curves and log$_{10}$(Emax/EC$_{50}$) following stimulation with the indicated concentrations of exendin-4 (Ex-4) in INS-1 832/3 GLP-1R KO cells transiently transfected with GLP-1R-SmBiT WT or V229A and LgBiT-mini-$G_s$; n=5. (**B**) Mini-$G_q$ recruitment dose response curves and log$_{10}$(Emax/EC$_{50}$) after stimulation with the indicated concentrations of Ex-4 in INS-1 832/3 GLP-1R KO cells transiently transfected with GLP-1R-SmBiT WT or V229A and LgBiT-mini-$G_q$; n=6. (**C**) β-arrestin 2 (βarr2) recruitment dose response curves and log$_{10}$(Emax/EC$_{50}$) after stimulation with the indicated concentrations of Ex-4 in INS-1 832/3 GLP-1R KO cells transiently transfected with GLP-1R-SmBiT WT or V229A and LgBiT-βarr2; n=5. (**D**) Mini-$G_s$ over βarr2 bias calculation for GLP-1R V229A *vs* WT. (**E**) GLP-1R WT *vs* V229A plasma membrane activation after stimulation with 100 nM Ex-4 in INS-1 832/3 GLP-1R KO cells co-transfected with Nb37-SmBiT, LgBiT-CAAX and SNAP/FLAG-hGLP-1R WT or V229A, measured by NanoBiT bystander complementation assay; area under the curve (AUC) also shown; n=6. (**F**) As in (**E**) but for GLP-1R WT *vs* V229A endosomal activation in INS-1 832/3 GLP-1R KO cells co-transfected with Nb37-SmBiT, Endofin-LgBiT and SNAP/FLAG-hGLP-1R WT or V229A; n=6. Data is mean +/- SEM; ns, non-significant, *p<0.05, **p<0.01 by paired t-test or two-way ANOVA with Sidak's multiple comparison test.

The online version of this article includes the following figure supplement(s) for figure 7:

**Figure supplement 1.** Schematic of the Nb37 bystander NanoBiT complementation assay.

---

(*Figure 8H, I*). Loading of transduced islets with excess cholesterol led to a loss of secretory responses to exendin-4 for both WT and V229A mutant GLP-1R (*Figure 8—figure supplement 1A, B*).

## Discussion

In this study, we have focused on assessing the potential for modulation of cholesterol interactions as a strategy to control GLP-1R responses in pancreatic beta cells. We have first shown that exogenous cholesterol increases via prolonged dietary changes in mice lead to a decrease in the capacity of the GLP-1R for in vivo glucoregulation, without affecting glucose levels under vehicle conditions. Further investigations showed that GLP-1R-mediated cAMP responses were reduced in islets extracted from these mice, suggesting that a high cholesterol diet specifically impacts on beta cell GLP-1R activity. Similarly, acute cholesterol increases in mouse islets exposed to cholesterol-loaded MβCD decreased the GLP-1R capacity to potentiate insulin secretion, validating our observed in vivo effects.

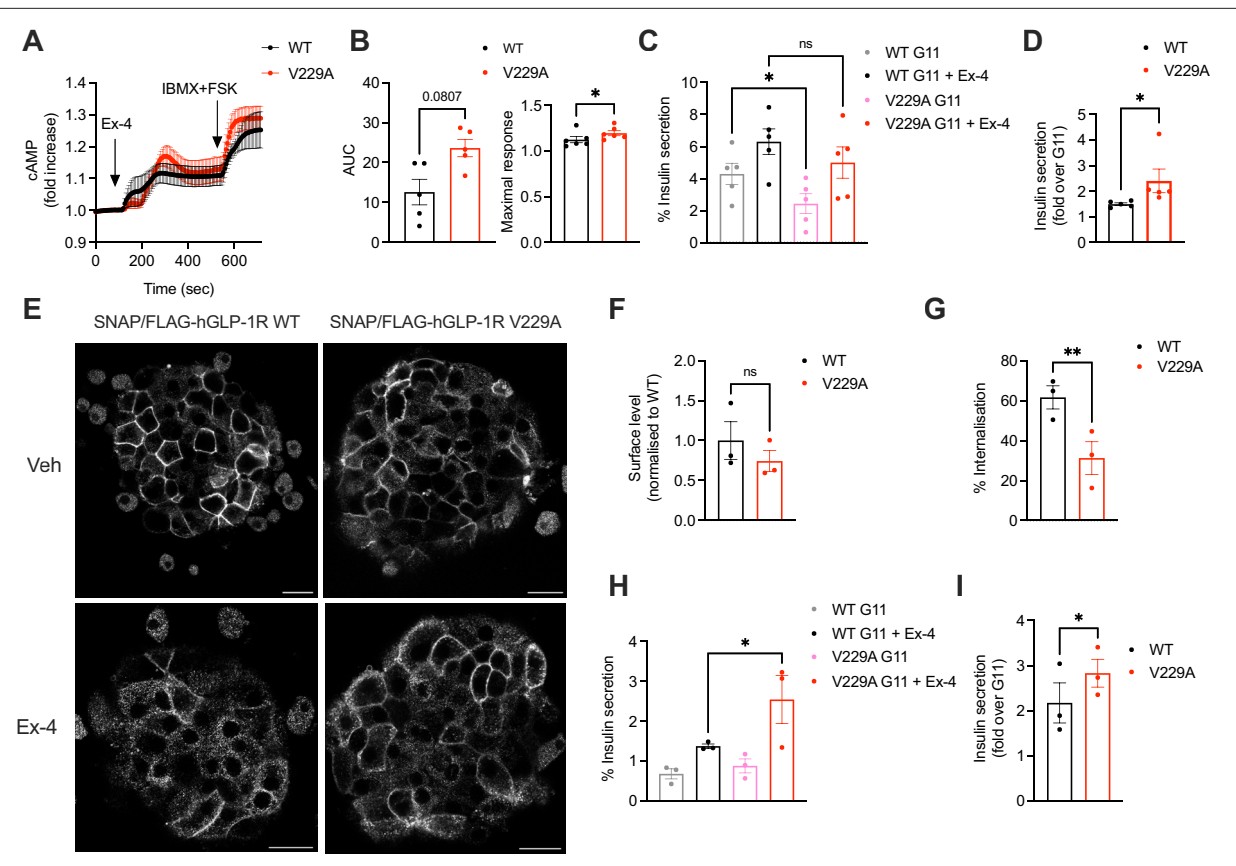

**Figure 8.** Functional responses of glucagon-like peptide-1 receptor (GLP-1R) wild-type (WT) *vs* V229A in pancreatic beta cells and primary islets. (**A**) cAMP responses of INS-1 832/3 SNAP/FLAG-hGLP-1R WT *vs* V229A cells transduced with the Green Up Global cAMP cADDis biosensor before stimulation with 100 nM Exendin-4 (Ex-4) followed by 100 μM isobutyl methylxanthine (IBMX) +10 μM forskolin (FSK) for maximal response. (**B**) AUC and maximal response for the Ex-4 period from (**A**); n=5. (**C**) Insulin secretion from INS-1 832/3 SNAP/FLAG-hGLP-1R WT *vs* V229A cells following stimulation with 11 mM glucose (**G11**) +/-100 nM Ex-4; n=5. (**D**) Insulin secretion Ex-4-fold increase over G11 calculated from data in (**C**). (**E**) Representative images of GLP-1R KO mouse islets transduced with adenoviruses expressing SNAP/FLAG-hGLP-1R WT or V229A, labelled with SNAP-Surface Alexa Fluor 647 prior to stimulation with vehicle (Veh) or 100 nM Ex-4 for 5 min; size bars, 100 μm. (**F**) Surface expression of SNAP/FLAG-hGLP-1R WT *vs* V229A expressed in GLP-1R KO mouse islets from (**E**); n=3. (**G**) Percentage of SNAP/FLAG-hGLP-1R WT *vs* V229A internalisation in GLP-1R KO mouse islets following stimulation with 100 nM Ex-4 for 5 min; n=3. (**H**) Insulin secretion responses from GLP-1R KO islets transduced with SNAP/FLAG-hGLP-1R WT *vs* V229A adenoviruses following stimulation with G11 +/-100 nM Ex-4; n=3. (**I**) Insulin secretion Ex-4-fold increase over G11 calculated from data in (**H**). Data is mean +/- SEM; ns, non-significant, *p<0.05, **p<0.01 by paired t-test or one-way ANOVA with Sidak's multiple comparison test.

The online version of this article includes the following figure supplement(s) for figure 8:

**Figure supplement 1.** Insulin secretion of glucagon-like peptide-1 receptor (GLP-1R) wild-type (WT) *vs* V229A-expressing islets under high cholesterol conditions.

In agreement with previous reports (*Asalla et al., 2016*), acute (1 hr) beta cell exposure to high cholesterol leads to increased insulin secretion under 11 mM glucose conditions, but the GLP-1R is unable to further potentiate secretion to the same extent than with unmodified cholesterol levels.

While we have observed a direct detrimental effect of a cholesterol-enriched diet on islet GLP-1R cAMP outputs, we cannot rule out that the reduced receptor capacity to lower blood glucose levels following cholesterol loading might encompass further negative effects of high cholesterol on GLP-1R downstream signalling. Persistently elevated mouse islet cholesterol levels have been shown to cause a decrease in the expression of cholesterol synthesis genes (*Hmgcr*) and increased expression of steroidogenic acute and regulatory protein (StAR), while also decreasing glucose-stimulated insulin secretion (*Akter et al., 2022*). StAR is a steroid biogenesis factor involved in cholesterol transport from the outer to the inner mitochondrial membrane (*Manna et al., 2016*) expressed at low levels in beta cells (*Hogan et al., 2019b*), with increased StAR expression linked to reduced beta cell mitochondrial function (*Hogan et al., 2019a*). We (*Gregory Austin et al., 2024*), and others (*Kang et al.,*

*2015*), have shown that the GLP-1R plays an important role in the control of beta cell mitochondrial function, a key process for optimal insulin secretion (*Ježek et al., 2022*), with our recent interactome results unveiling direct interactions between GLP-1R and cholesterol biosynthesis and metabolism factors located at ER-mitochondria membrane contact sites (*Gregory Austin et al., 2024*). It is, therefore, possible that high cholesterol might impact on the capacity of the GLP-1R to regulate mitochondrial and beta cell function via altered expression and interaction with cholesterol regulatory enzymes, hindering the insulin secretion potentiation effect of the receptor.

Conversely, in this study, we also found that a reduction in mouse islet cholesterol levels with simvastatin, which disrupts ER cholesterol synthesis by inhibiting HMGCR enzymatic activity (*Röhrl and Stangl, 2018*; *Friesen and Rodwell, 2004*; *Plosker and Simvastatin, 1995*), increased exendin-4-mediated cAMP and insulin secretion responses. While this partial reduction in cholesterol levels appears beneficial for beta cell GLP-1R function, we previously observed that a much more drastic cholesterol extraction using MβCD leads to the opposite effect, inhibiting agonist-induced GLP-1R cAMP responses (*Buenaventura et al., 2019*), a result that we hypothesise is linked to the profound disruption of plasma membrane architecture caused by MβCD (*Mahammad and Parmryd, 2008*). The use of statins at higher concentrations than in the present study has been previously associated with varying effects on insulin secretion: simvastatin was shown to reduce, while pravastatin increased insulin secretion in response to glucose, with GLP-1 or exendin-4 exposure restoring secretory responses from statin-treated cells (*Yaluri et al., 2015*). Statin use has been associated with increased risk of T2D by poorly understood mechanisms involving both reduced insulin sensitivity and impaired beta cell insulin secretion (*Laakso and Fernandes Silva, 2023*), the latter ameliorated by GLP-1R agonist exposure (*Buldak et al., 2022*). Considering our results as well as previous literature, we speculate that patients with increased cholesterol levels might also exhibit reduced beta cell responses to incretin therapies unless dyslipidaemia is subsequently ameliorated following weight loss. In addition, patients taking cholesterol modifying drugs like statins are particularly good candidates for incretin use as they might present with improved incretin responses that would also help prevent a potentially increased risk of developing T2D in this patient subset. Further research to determine responses to GLP-1R agonists in individuals with different cholesterol levels before and after treatment with cholesterol lowering agents will be required to elucidate this matter.

The GLP-1R, as an integral membrane protein and 7-TMD GPCR, tightly interacts with its lipid-rich plasma membrane microenvironment (*Corradi et al., 2018*) and is also known to segregate to cholesterol-rich, flotillin-positive lipid nanodomains upon activation (*Buenaventura et al., 2019*). It is, therefore, plausible that direct interaction of cholesterol molecules with specific binding pockets in the GLP-1R might at least be partially responsible for cholesterol modulation of GLP-1R function. To test this possibility, we performed cgMD simulations of GLP-1R in active *vs* inactive conformations in a model mammalian plasma membrane using the latest Martini 3 forcefield and cholesterol parameters, identifying specific receptor sites with high cholesterol occupancies and residence times. The fact that the highest cholesterol occupancy residues identified correspond to either ring-containing (Phe) or branched (Val, Leu, Ile) amino acids suggests that these residue types might play important roles in cholesterol stabilisation at GLP-1R binding pockets. Importantly, the GLP-1R-cholesterol binding sites identified here do not overlap with previously predicted cholesterol consensus motifs (CCM) (*Hanson et al., 2008*) or Cholesterol Recognition Amino Acid Consensus (CRAC) motifs (*Jafurulla et al., 2011*) identified for other GPCRs, confirming the theory that GPCRs lack predictable primary sequence consensus motifs for cholesterol binding (*Zhang et al., 2020*; *Taghon et al., 2021*). The GLP-1R-cholesterol binding sites also presented notable differences between active and inactive receptor conformations, suggesting that cholesterol binding is regulated by agonist binding and receptor activation. GLP-1R in its inactive apo-state is constantly moving but prefers to adopt a closed ECD conformation in absence of peptide binding (*Wu et al., 2020*). Active, peptide-bound receptors, on the other hand, adopt a more open conformation resulting from the peptide agonist interacting with the ECD, leading to prominent changes at TMD regions which stabilise G protein binding and activation (*Zhang et al., 2017*; *Zhang et al., 2020*; *Liang et al., 2018*). The conformational shift between the receptor in active and inactive states is, therefore, predicted to affect GLP-1R-cholesterol interactions, thereby leading to distinct active *vs* inactive cholesterol binding sites. Interestingly, one of the most prominent GLP-1R-cholesterol binding sites identified

here lies within the functionally important TM5-TM6 region required for G protein signal transduction, indicating the potential impact of cholesterol binding on GLP-1R activation and downstream signalling.

To investigate the importance of direct cholesterol interactions in modulating beta cell GLP-1R function, we have screened 12 residues selected from the cholesterol binding sites identified in our cgMD simulations, covering a wide range of occupancy, residence time, ability to tolerate the planned mutation, and varied locations within binding sites in active *vs* inactive receptor states. Interestingly, the residues that had significant effects on GLP-1R internalisation while mutated to alanine were all non-polar branched amino acids, with some of these also predicted to cause changes in the cavity in which they reside. Specifically, changing the Val229 residue to alanine, selected in this study for further investigation, was predicted to trigger a change in the cavity and a buried exposure switch at the inactive receptor, thereby potentially modifying the interactions between itself and other pocket residues with cholesterol.

Further detailed investigations of GLP-1R V229A behaviours in beta cells revealed an overall reduction in cholesterol binding under vehicle conditions, which correlated with reduced lateral diffusion, average displacement, and speed of movement across the plasma membrane of the V229A receptor under these conditions. Detailed examination of the receptor oligomerisation state using N&B also demonstrated a greater population of higher order oligomers under vehicle conditions for the V229A *vs* WT GLP-1R. RICS lateral diffusion takes into consideration the movement of molecules in regions of interest within cells *Rossow et al., 2010*; the reduced lateral diffusion of GLP-1R V229A could, therefore, be due to the mutant receptor existing as a subpopulation of pre-clustered, higher order oligomeric assemblies in absence of agonist stimulation, predicted to cause a decrease in the movement of receptors within these clusters. Interestingly, we detected a similar pattern of reduced lateral diffusion of the WT GLP-1R under vehicle conditions following cholesterol lowering with simvastatin, while increased cholesterol led to a reduction in the capacity of exendin-4 to slow down the GLP-1R, suggesting that plasma membrane cholesterol levels, and therefore receptor-cholesterol interactions, are key for the regulation of this behaviour, which in turn determines the capacity of receptors to segregate to lipid nanodomains, modulating downstream signalling responses (*Jones et al., 2020*). Accordingly, GLP-1R V229A presented with a higher tendency to segregate to flotillin-positive lipid nanodomains, or lipid rafts, in the apo-state, so that the increased clustering of GLP-1R V229A could potentially be triggered by increased mutant receptor partitioning to these nanodomains. This observation also highlights the complexity that modifying a residue within a cholesterol binding site might have on the overall pattern of receptor interactions with cholesterol and other plasma membrane lipids, as increased segregation to these cholesterol-rich lipid rafts occurs despite the overall reduction in total cholesterol binding under vehicle conditions observed for the V229A mutant receptor.

GLP-1R V229A receptor pre-clustering under vehicle conditions did not trigger any measurable changes in the lateral diffusion of plasma membrane lipid-ordered nanodomains measured as a whole using the lipid dye Laurdan, which allows for the detection of different plasma membrane organisation states as either liquid ordered or disordered (*Gunther et al., 2021*). Laurdan is a non-species-specific lipid probe that cannot discriminate between different lipid nanodomains at the plasma membrane, hence potentially explaining the lack of effect on their overall lateral diffusion by expression of the mutant *vs* WT GLP-1R. We nevertheless observed, by RICCS analysis, a tendency for reduced co-diffusion of these highly packed lipid nanodomains with GLP-1R V229A under vehicle conditions, an effect that is only present in stimulated conditions for the WT receptor, with no further change after agonist stimulation observed for the V229A mutant, again suggesting a different pattern of interaction of V229A *vs* WT GLP-1R with these plasma membrane lipid nanodomains.

Underscoring the importance that direct cholesterol engagement has in the modulation of GLP-1R function, the changes in cholesterol binding and plasma membrane behaviours introduced by the V229A substitution led to significant effects on GLP-1R trafficking and signalling in beta cells: V229A caused a decrease in GLP-1R recruitment to CCPs and receptor internalisation, accompanied by an increase in $G\alpha_s$ (measured with mini-$G_s$) and β-arrestin 2 engagement, as well as increased plasma membrane recycling following exendin-4 stimulation. Overall, the V229A receptor presented with biased signalling favouring $G\alpha_s$ coupling over β-arrestin 2 recruitment and receptor localisation and activation at the plasma membrane. GLP-1R biased signalling also occurs in WT receptors following activation with modified peptide agonists (*Jones, 2022*). For example, biased GLP-1R signalling

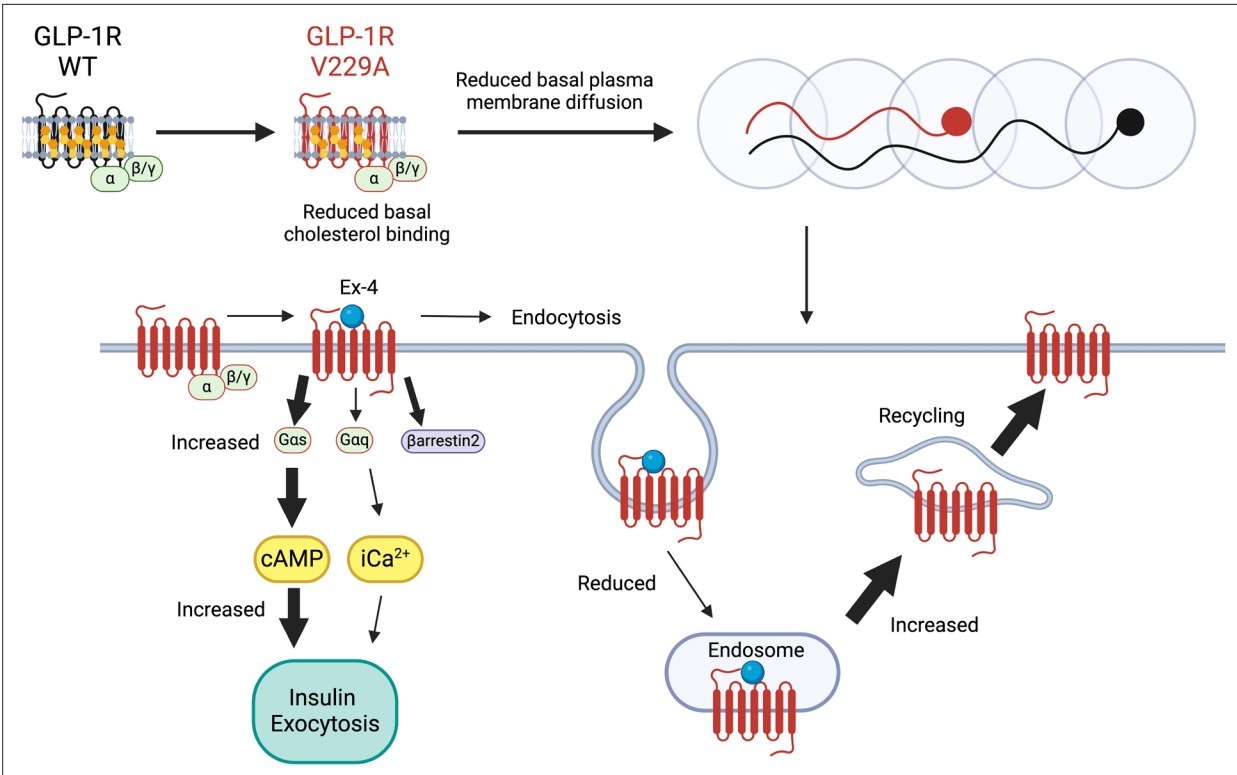

**Figure 9.** Schematic diagram of effects of cholesterol binding mutant glucagon-like peptide-1 receptor (GLP-1R) V229A on GLP-1R function. Thick arrows indicate increased and thin arrows decreased pathway engagement.

is observed following stimulation with exendin-phe1, which carries a phenylalanine substitution at position 1 of exendin-4, causing a similar reduction in receptor internalisation and increased plasma membrane recycling to GLP-1R V229A (*Jones et al., 2018*). In contrast, however, exendin-phe1 triggers essentially no β-arrestin 2 recruitment to the receptor, as well as prolonged GLP-1R-induced cAMP and insulin secretion (*Jones et al., 2018*), together with reduced GLP-1R clustering and lipid raft recruitment (*Buenaventura et al., 2019*), while GLP-1R V229A presents with increased oligomerisation at vehicle conditions and acute effects on beta cell function (summarised in *Figure 9* and *Table 1*). Some of these effects are, as previously highlighted, analogous to those observed by reducing beta cell/islet cholesterol with simvastatin: in addition to a reduction in GLP-1R plasma membrane lateral diffusion under vehicle conditions, both manipulations led to acute increases in GLP-1R-induced cAMP and islet insulin secretion responses. While the V229A mutation did not confer an advantage to ameliorate insulin secretion responses to exendin-4 in islets loaded with excess cholesterol ex vivo, it remains to be studied if this or another modification in GLP-1R cholesterol binding sites can improve in vivo responses to incretin therapies under diet-induced metabolic stress conditions. Regardless, when taken together, the results of this study indicate that modulation of direct GLP-1R-cholesterol interactions is a valid strategy to fine-tune beta cell incretin receptor responses, providing a new avenue for further investigation of novel ways to improve GLP-1R signalling outputs.

The phenomenon of cholesterol modification of GPCR outputs by direct receptor binding has been previously demonstrated for other GPCR classes, especially class A GPCRs (*Moreau et al., 2023*; *Jakubík and El-Fakahany, 2021*). For example, cholesterol binding sites were identified in the adenosine $A_{2A}$ receptor; mutation of residues within the predicted receptor-cholesterol binding sites caused changes in basal receptor activity, ligand binding, G protein coupling, and cAMP responses (*McGraw et al., 2022*). Receptor-cholesterol interactions were also affected by either agonist or antagonist binding (*Moreau et al., 2023*). Similarly, determining the effect that different agonists and/or antagonists might have on GLP-1R-cholesterol interactions might also provide further valuable insights into their specific mechanisms of action. Additionally, it would be highly pertinent to study the effects of modifying GLP-1R cholesterol binding sites in other cell types that express the receptor

Table 1. Overview of effects of glucagon-like peptide-1 receptor (GLP-1R) V229A in inactive and active states compared with GLP-1R wild-type (WT).
(↑ increased, ↓ decreased, ≈ unchanged).

| | GLP-1R V229A vs GLP-1R WT | |
| --- | --- | --- |
| Assay | Vehicle (inactive) | Exendin-4 (active) |
| Surface expression | ≈ (↓) | N/A |
| Cholesterol binding | ↓ | ≈ |
| Plasma membrane diffusion | ↓↓ | ≈ |
| Plasma membrane displacement | ↓ | ≈ |
| Plasma membrane speed | ↓ | ≈ |
| Oligomerisation | ↑ | ≈ |
| Lipid raft recruitment | ↑ | ↑ |
| CCP recruitment | ≈ | ↓ |
| Internalisation | N/A | ↓ |
| Recycling | N/A | ↑ |
| Degradation | N/A | ≈ |
| Mini-G$_s$ recruitment | N/A | ↑↑ |
| β-arrestin 2 recruitment | N/A | ↑ |
| Mini-G$_q$ recruitment | N/A | ≈ |
| Activation at plasma membrane | N/A | ↑↑ |
| Activation at endosomes | N/A | ↑ |
| cAMP | N/A | ↑ |
| Insulin secretion | ↓ cells / ≈ islets | ≈ cells / ↑ islets |

beyond pancreatic beta cells/islets, specifically in neurons where the receptor plays important roles in the control of satiation and nausea (*Trapp and Brierley, 2022*), which might be modulated by modified receptor-cholesterol interactions.

Our results serve as a first for a class B1 GPCR, highlighting the importance of GPCR-lipid interactions on receptor function and dynamics for this receptor class. It further confirms cholesterol binding pockets as high value sites for GLP-1R allosteric modulation, with the capacity to fine-tune GLP-1R trafficking, signalling, and functional outputs. Further investigations to fully elucidate the effect of the V229A substitution on GLP-1R-cholesterol interactions might include detailed atomistic MD simulations of WT *vs* V229A receptors, as well as resolving cryo-EM structures of mutant and WT receptors in nanodiscs containing relevant lipid compositions to preserve receptor-lipid interactions in their native environment. Large cgMD simulations including increased numbers of WT or mutant receptors in a membrane environment with varying cholesterol levels could also be performed to determine the effect of cholesterol on GLP-1R oligomerisation. Finally, further high-throughput investigations to evaluate all other key residues involved in GLP-1R-cholesterol binding will generate high value data for the future development of novel GLP-1R allosteric modulators with improved signalling properties for the treatment of T2D and obesity.

# Materials and methods
## Animal studies
In vivo studies were carried out under the approval of the UK Home Office Animals (Scientific Procedures) Act 1986 Project Licence PP7151519 (PPL Holder: Dr A. Martinez-Sanchez, Imperial College London) at the Central Biological Services unit of Imperial College London. Animals were housed in individually ventilated cages in groups up to five adult mice in a controlled environment (temperature,

21 to 23°C; 12 hr light and 12 hr dark cycles) with free access to standard chow (*Bitsi et al., 2023*) or high-cholesterol diet when relevant.

## Intraperitoneal glucose tolerance tests (IPGTTs)

C57BL/6 J (RRID:MGI:3028467) WT mice were fed a standard chow diet or a rodent diet with 10 kcal% fat and 2% cholesterol (high cholesterol diet; D01101902CR, Research Diets, Inc) for 5 wk (8–9 female mice per condition). Animals were fasted for 2 hr prior to intraperitoneal injection of glucose (2 g/kg)+/- exendin-4 (1 nmol/kg) and continued to fast for 6 hr when blood glucose levels were assessed from the tail vein using a Contour glucometer (Bayer) and strips at 0-, 10-, 30-, and 60 min time-points, with glucose response curves generated from these data.

## Mouse pancreatic islet isolation and culture

Primary islets from mixed sex backgrounds were isolated from C57BL/6 J WT or GLP-1R KO mice (the latter generated *in house* by CRISPR/Cas9 deletion). Pancreata were infused post-mortem with RPMI-1640 medium, (21875091, Thermo Fisher Scientific) containing 1 mg/mL collagenase (*Clostridium histolyticum*) (S1745602, Nordmark Biochemicals), dissected, and incubated in a 37 °C water bath for 10 min to digest the tissue. RPMI-1640 medium with 10% foetal bovine serum (FBS) (F7524, Sigma Aldrich) was added and mixed vigorously for 20 s to stop the reaction. The dissected pancreata were subsequently washed in RPMI-1640 medium prior to islet separation using a gradient with Histo-paque-1119 and −1083 (both from Sigma Aldrich). Isolated islets were transferred to non-adherent dishes containing RPMI-1640 medium with 2 mM L-glutamine and 11 mM D-glucose supplemented with 10% FBS and 1% penicillin/streptomycin (p/s) (P4458, Sigma Aldrich) in a humidified incubator at 37 °C and 5% $CO_2$. Islets were allowed to recover for 24 hr prior to use (*Bitsi et al., 2023*).

## Cholesterol modification treatments

To increase cholesterol, islets isolated from chow-fed C57BL/6 J WT mice were pre-incubated in 20 mM cholesterol loaded onto methyl-β-cyclodextrin (MβCD) (C4951, Sigma Aldrich) diluted in 1 X Krebs-Ringer bicarbonate-Hepes (KRBH) buffer [140 mM NaCl, 3.6 mM KCl, 1.5 mM $CaCl_2$, 0.5 mM $MgSO_4$, 0.5 mM $NaH_2PO_4$, 2 mM $NaHCO_3$, 10 mM Hepes, saturated with 95% $O_2$/5% $CO_2$; pH 7.4 and 0.1% bovine serum albumin (BSA)] for 1 hr at 37 °C prior to assays being carried out. Control islets were pre-incubated in KRBH buffer alone. For reduced cholesterol, islets were pre-incubated in 10 µM simvastatin (S6196, Sigma Aldrich) diluted in RPMI-1640 supplemented with 10% lipoprotein deficient serum (LPDS) (S5394, Sigma Aldrich) and 1% p/s for a minimum of 18 hr prior to the assays. Control islets were pre-incubated in RPMI-1640 supplemented with 10% FBS and 1% p/s for the same time.

## Assessment of cholesterol levels

### By fluorescence labelling

Islets were washed once with PBS and cholesterol labelled according to the Cholesterol Assay kit (cell-based) (ab133116, Abcam). Briefly, islets were fixed for 15 min with the cell-based assay Fixative solution. Islets were then washed with the Cholesterol Detection wash buffer, incubated for a minimum of 1 hr in Filipin III diluted in the Cholesterol Detection Assay buffer, and washed again in wash buffer before immediate imaging on a Zeiss LSM-780 inverted confocal laser-scanning microscope from the Facility for Imaging by Light Microscopy (FILM) at Imperial College London, with a 63 X objective and a laser excitation wavelength of 340–380 nm and emission of 385–470 nm. Raw fluorescence images were analysed using Fiji to determine the average fluorescent intensity representing the level of cholesterol per islet.

### By Amplex Red assay

Around 18–20 islets isolated from mice fed either a standard chow diet or a high cholesterol diet for 5 wk were resuspended in a chloroform:methanol mixture (2:1 ratio) and centrifuged in a SpeedVac to extract the lipids containing cholesterol in a thin film. The film was then resuspended in 1 X Amplex Red Cholesterol kit (A12216, Thermo Fisher Scientific) reaction buffer supplemented with 0.1% Triton X-100 and used to determine the cholesterol content according to the manufacturer's instructions. Cholesterol level was further normalised to corresponding protein concentration determined using

the Pierce BCA Protein Assay Kit (23227, Thermo Fisher Scientific) following the manufacturer's instructions.

## Cell culture

INS-1 832/3 cells (RRID:CVCL_ZL55), a rat insulinoma subline with preserved incretin responses (*Hohmeier et al., 2000*), and derivative INS-1 832/3 cells with endogenous rat GLP-1R deleted by CRISPR/Cas9 [INS-1 832/3 GLP-1R KO (*Naylor et al., 2016*)], were grown in RPMI-1640 medium with 2 mM L-glutamine and 11 mM D-glucose, supplemented with 10% FBS, 1% p/s, 10 mM HEPES solution (H3537, Sigma-Aldrich), 1 mM sodium pyruvate (11360070, Thermo Fisher Scientific) and 0.05 mM β-mercaptoethanol (M3148, Sigma Aldrich) in a humidified incubator at 37 °C and 5% $CO_2$. INS-1 832/3 SNAP/FLAG-hGLP-1R WT and V229A cells were generated in house from INS-1 832/3 GLP-1R KO cells following transfection with pSNAP/FLAG-hGLP-1R (Cisbio) or pSNAP/FLAG-hGLP-1R-V229A (generated *in house* by site-directed mutagenesis, see below), followed by selection with 1 mg/mL G418 sulphate (A1720, Sigma Aldrich). Cells were labelled with SNAP-Surface Alexa Fluor 546 (New England Biolabs) and SNAP-expressing fluorescent cells sorted by fluorescence-activated cell sorting (FACS) and maintained as above with media supplemented with 0.5 mg/mL G418. Myco-plasma testing was performed monthly.

## cAMP live imaging
### cAMP cADDis biosensor imaging

INS-1 832/3 SNAP/FLAG-hGLP-1R WT *vs* V229A cells seeded onto glass bottom MatTek dishes, or pancreatic islets, were transduced overnight with the Green Up cADDis biosensor in a BacMam vector (#U0200G, Montana Molecular), according to the manufacturer's instructions. Cells were washed and imaged in RPMI without phenol red, while islets were washed and encased in Matrigel (Corning) and imaged in KRBH buffer supplemented with 6 mM glucose. Green fluorescence was recorded at 6 s intervals in a 20 X objective 37 °C heated stage Nikon spinning disk field scanning confocal microscope. A baseline reading was taken for 1 min, then 100 nM exendin-4 was added and cells/islets imaged for a further 5 min, prior to addition of 100 µM isobutyl methylxanthine (IBMX) +10 µM forskolin (FSK) and 2 min imaging for maximal response. Fluorescent intensity per islet or cell was calculated using Fiji and responses plotted relative to the baseline period. Area under the curve (AUC) was calculated for the exendin-4 period using GraphPad Prism 10.

### cAMP $^{T}EPAC^{VV}$ fluorescence resonance energy transfer (FRET) imaging

Islets extracted from Pdx1$^{Cre-ERT}$-CAMPER mice, conditionally expressing the $^{T}EPAC^{VV}$ cAMP biosensor in beta cells under the control of the *Pdx1* promoter conjugated to a mutant oestrogen receptor sequence (*Bitsi et al., 2023*), were incubated in 1 µM 4-hydroxytamoxifen for 24 hr to induce $^{T}EPAC^{VV}$ expression, treated with simvastatin as above, washed, encased in Matrigel and imaged in 6 mM glucose KRBH buffer using a Zeiss LSM-780 inverted confocal laser-scanning microscope in a 20 X objective at 37 °C. FRET between the Turquoise donor and the Venus acceptor within $^{T}EPAC^{VV}$ was recorded every 6 s. A baseline reading was taken for 1 min, followed by a 5 min recording period with 100 nM exendin-4. Glucose was then increased to 16 mM and islets imaged for a further 3 min before addition of 100 µM IBMX +10 µM FSK as above to record maximal responses. Fluorescent intensities for Turquoise and Venus channels were extracted per islet in Fiji and Turquoise/Venus ratio calculated corresponding to cAMP levels. Responses were presented as average intensity traces per islet relative to its baseline period, and AUC calculated as above.

## Insulin secretion assays
### Insulin secretions in islets

Purified islets were incubated in 3 mM glucose KRBH buffer for 1 hr in a shaking water bath at 37 °C. Islets were then incubated in 11 mM glucose KRBH buffer +/-100 nM exendin-4 or stimulated with a secretagogue cocktail (20 mM glucose, 30 mM KCl, 10 µM FSK and 100 µM IBMX) for 30 min. Super-natants containing secreted insulin were collected, spun at 1000 rpm for 1 min, and transferred to new Eppendorf tubes. Islets were then lysed using acidic ethanol (75% ethanol, 15 mM HCl), sonicated 3X 10 s, and centrifuged at 1000 rpm for 3 min, and supernatants collected to assess internal insulin content and stored at –20 °C. Each condition was analysed in triplicate with 10 islets per replicate.

## Insulin secretions in cells

INS-1 832/3 SNAP/FLAG-hGLP-1R WT or V229A cells were seeded onto 48-well plates, pre-incubated in 3 mM glucose media overnight, washed with KRBH buffer and incubated in 3 mM glucose KRBH buffer for 1 hr before stimulation with 11 mM glucose KRBH +/-100 nM exendin-4 for 1 hr. Supernatants containing secreted insulin were collected and spun at 1000 rpm for 3 min and stored at –20 °C. Cells were lysed with lysis buffer (KRBH supplemented with 0.1% BSA and 1% Triton X-100), collected in an Eppendorf tube, sonicated 3X 10 s, and centrifuged at 13,000 rpm for 10 min. Supernatants containing internal insulin were collected to determine total insulin content. Samples were stored at –20 °C until they were analysed.

Insulin concentrations from cells and islets were determined using the Insulin Ultra-Sensitive HTRF Assay Kit (2IN2PEG, Cisbio) according to the manufacturer's instructions. Standard curves were generated using GraphPad Prism 10 and insulin concentration for each sample interpolated from the standard curve. Percentage of insulin secreted was determined from the secreted over the total (internal +secreted) insulin concentrations (see *Supplementary file 3* for raw insulin data included in this study).

## Coarse-grained molecular dynamics (cgMD) simulations

The structure of GLP-1R in inactive form was adopted from GPCRdb (*Pándy-Szekeres et al., 2023*), using refined GLP1R model PDB ID: 6LN2 (*Wu et al., 2020*; *Zhang et al., 2021*). The active form was adopted from cryoEM structure PDB ID: 6x18 (*Zhang et al., 2020*) with the missing loop modelled back in using MODELLER version 10.4 (*Eswar et al., 2006*). The GLP-1R was coarse-grained using Martinize2 (*Kroon et al., 2022*) with MARTINI 3 forcefield (*Souza et al., 2021*). The ElNeDyn elastic network restraint was applied to active and inactive GLP-1R using an elastic bond force constant of 500 kJ/mol/nm$^2$ and an upper cut-off of 0.9 nm (*Periole et al., 2009*). The transmembrane region of GLP-1R was predicted using PPM 3.0 Web Server (*Lomize et al., 2022*) and embedded into an asymmetric lipid membrane bilayer in a 18 × 18 × 17 nm$^3$ box using insane.py (*Wassenaar et al., 2015*), with lipid composition as follows: POPC (30%), DOPC (30%), POPE (8%), DOPE (7%) and cholesterol (25%) in the outer leaflet and POPC (5%), DOPC (5%), POPE (20%), DOPE (20%), POPS (8%), DOPS (7%), PIP$_2$ (10%), and cholesterol (10%) in the inner leaflet. Mammalian plasma membrane composition was adopted from *Song et al., 2019*. The new MARTINI 3 lipid model for cholesterol was used in this study (*Borges-Araújo et al., 2022*; *Borges-Araújo et al., 2023*). Each system was solvated using MARTINI water (*Souza et al., 2021*) and 150 nM NaCl, followed by the standard minimisation and equilibration protocols from CHARMM-GUI Martini Marker (*Qi et al., 2015*). The production simulation was 10 microseconds in length and was repeated 3 X. The v-rescale thermostat (tau 1.0 picoseconds) (*Bussi et al., 2007*) and the Parrinello–Rahman barostat (tau 12.0 picoseconds) (*Parrinello and Rahman, 1981*) were used to maintain temperature (303.15 K) and pressure (1 bar) on all production runs. All simulations were done using GROMACS 2022.4 (*Abraham et al., 2015*). Cholesterol interaction profiles were calculated using PyLipID (*Song et al., 2022*) with a cut-off of 0.7 nm. Trajectory was analysed using gromacs tool and VMD (*Humphrey et al., 1996*).

## Site-directed mutagenesis of GLP-1R cholesterol binding sites

Human SNAP/FLAG-tagged GLP-1R (SNAP/FLAG-hGLP-1R) (Cisbio) was used as the template to generate GLP-1R cholesterol binding mutants for screening. Site-directed mutagenesis primers (*Table 2*) were designed to replace the relevant amino acid residue with alanine and carried out using the PfuUltra II Fusion HS DNA Polymerase kit (600670, Agilent) according to the manufacturer's instructions. PCR products were treated with 1 µL Dpn1 (10 U/µL) for 1 hr at 37 °C to digest parental DNA. Digested DNA samples were transformed into NEB10-beta Competent *E. coli* High Efficiency cells (C3019H, New England Biolabs) according to the manufacturer's instructions. Mutant plasmids were sent for Sanger sequencing (Azenta/Genewiz) to confirm the amino acid change to alanine.

## GLP-1R surface expression and internalisation by confocal microscopy

INS-1 832/3 GLP-1R KO cells were transiently transfected with SNAP/FLAG-hGLP-1R WT or cholesterol-binding mutants; purified GLP-1R KO mouse islets were transduced with SNAP/FLAG-hGLP-1R WT *vs* V229A adenoviruses (generated by VectorBuilder) at an MOI of 1. Cells or islets were incubated for 24 hr post-transfection/transduction before labelling with 1 µM SNAP-Surface

**Table 2.** Site-directed mutagenesis primers for SNAP/FLAG-hGLP-1R cholesterol binding mutant generation.

| GLP-1R cholesterol binding mutant | Forward primer (5'–3') | Reverse primer (5'–3') | Side chain charge change |
|---|---|---|---|
| **BATCH 1 SCREEN** | | | |
| I323A | CATTGGGGTGAACTTCCTCGCCTTTGTTCGGGTCATCTGC | GCAGATGACCCGAACAAAGGCGAGGAAGTTCACCCCAATG | non-polar to non-polar |
| S163A | CTCTGGTTATCGCCGCTGCGATCCTCCTC | GAGGAGGATCGCAGCGGCGATAACCAGAG | polar to non-polar |
| S225A | CCAGGACTCTCTGCCTGCCGCCTGGTG | CACCAGGCGGCAGGCCAGAGAGTCCTGG | polar to non-polar |
| V229A | GAGCTGCCGCCTGGCGTTTCTGCTCATGC | GCATGAGCAGAAACGCCAGGCGGCAGCTC | non-polar to non-polar |
| **BATCH 2 SCREEN** | | | |
| L167A | CCTCTGCGATCCTCGCCGGCTTCAGACACC | GGTGTCTGAAGCCGGCGAGGATCGCAGAGG | non-polar to non-polar |
| V160A | CCTTCTCTGCTCGGCTATCGCCTCTGCGAT | ATCGCAGAGGCGATAGCCAGAGCAGAGAAGG | non-polar to non-polar |
| I165A | GTTATCGCCTCTGCGCGCCCTCCTCGGCTTCAG | CTGAAGCCGAGGAGGGCGCGCAGAGGCGATAAC | non-polar to non-polar |
| G151A | CATCTACACGGTGGCCTACGCACTCTCCT | AGGAGAGTGCGTAGGCCACCGTGTAGATG | non-polar to non-polar |
| **BATCH 3 SCREEN** | | | |
| D372A | GCCTTTGTGATGGCCGAGCACGCCCGG | CCGGGGCGTGCTCGGCCATCACAAAGGC | acidic to non-polar |
| H363A | CATCCCCCTGCTGGGGACTGCTGAGGTCATCTTTG | CAAAGATGACCTCAGCAGTCCCCAGCAGGGGGATG | basic to non-polar |
| L401A | AGGGGCTGATGGTGGCCATAGCATACTGCTTTGTCAACAAT | ATTGTTGACAAAGCAGTATGCTATGGCCACCATCAGCCCCT | non-polar to non-polar |
| I400A | CAGGGGCTGATGGTGGCCGCATTATACTGCTTTGTCAAC | GTTGACAAAGCAGTATAATGCGGCCACCATCAGCCCCTG | non-polar to non-polar |

fluorescent probes (New England Biolabs) for 15–20 min for cells, or 7 min for islets, in full media at 37°C. Cells were washed and imaged in RPMI-1640 without phenol red (32404014, Thermo Fisher Scientific), while islets were washed and imaged in 6 mM glucose KRBH buffer using a Nikon spinning disk field scanning confocal microscope with a 60 X oil objective at 37 °C. Cells were imaged by time-lapse microscopy with images taken every 6 s; a baseline recording was taken for 1 min before addition of 100 nM exendin-4 and further imaging for 10 min. Islets were imaged in vehicle conditions or 5 min after stimulation with 100 nM exendin-4 at 37 °C.

Raw images were analysed using Fiji and the full-width half maximum (FWHM) of a line profile macro, developed by Steven Rothery, FILM Facility, Imperial College London (https://www.imperial.ac.uk/medicine/facility-for-imaging-by-light-microscopy/software/fiji/), employed to determine the change in SNAP fluorescence intensity at the plasma membrane from individual cells. SNAP/FLAG-hGLP-1R surface expression levels were determined from the average intensity during the baseline period for cells or from the vehicle images for islets. Receptor internalisation was calculated from the loss of fluorescence intensity at the plasma membrane after 10 min of exendin-4 stimulation compared to baseline intensity for cells, or from the loss of fluorescence intensity after 5 min of exendin-4 stimulation vs vehicle for islets.

## PhotoClick cholesterol binding assay

PhotoClick cholesterol (37.5 nmoles, 700147 P, Avanti Polar Lipids) was agitated in 2 mM MβCD in Hanks Balanced Salt Solution (HBSS) (14025092, Thermo Fisher Scientific) overnight at room temperature to facilitate intracellular delivery. INS-1 832/3 SNAP/FLAG-hGLP-1R WT or V229A cells were seeded onto 6-well plates and incubated in 18.75 nmoles of PhotoClick cholesterol loaded onto MβCD in HBSS for 1 hr at 37 °C protected from light. Cells were then stimulated with 100 nM exendin-4 or vehicle for 2 min, washed with ice cold PBS and UV irradiated at 365 nm with a UV Crosslinker for 5 min in ice cold PBS to activate the photoreactive diazirine group in the PhotoClick cholesterol. Cells were then lysed in 1 X TBS (50 mM Tris-HCl, 150 mM NaCl, pH 7.4) supplemented with 1 mM EDTA, 1% Triton X-100, phosphatase inhibitor cocktail (P5726, Sigma Aldrich), and protease inhibitor cocktail (11873580001, Roche Diagnostics). Lysates were sonicated 3X 10 s, centrifuged at 10,000 rpm for 10 min, and incubated with anti-FLAG M2 affinity gel (A2220, Sigma Aldrich) rotating overnight at 4 °C to immunoprecipitate the SNAP/FLAG-hGLP-1R. Following immunoprecipitation, the bead-bound SNAP/FLAG-hGLP-1R was incubated with a click chemistry mix (20 μM rhodamine-azide, 1 mM TCEP, 100 μM TBTA, 1 mM CuSO$_4$) for 1 hr at room temperature gently agitating to fluorescently label the PhotoClick cholesterol. Urea buffer (2 X) was added in a 1:1 ratio and samples incubated for 10 min at 37 °C prior to 10% SDS-PAGE gel separation and imaging using a ChemiDoc MP imaging system to detect fluorescently labelled cholesterol. The proteins were then transferred onto a PVDF membrane and corresponding SNAP/FLAG-hGLP-1R levels detected by Western blotting with an anti-SNAP antibody (P9310S, New England Biolabs, RRID:AB_10631145). Data was represented as amount of bound Photo-Click cholesterol relative to SNAP/FLAG-hGLP-1R.

## Binding affinity assay

INS-1 832/3 GLP-1R KO, SNAP/FLAG-hGLP-1R WT, and V229A cells were seeded at 50,000 cells per well in a clear-bottom 96-well black plate coated with poly-d-Lysine. 24 hr after seeding, growth media was removed and cells were stimulated with fluorescent GLP-1R agonist exendin-asp3-TMR (0.1–1000 nM) in complete DMEM for 10 min at 37 °C. Cells were washed three times with PBS, then imaged in Krebs-Henseleit buffer using an EVOS M7000 cell imaging system with a Texas Red 2.0 light cube with $\lambda$ ex = 585 nm and $\lambda$ em = 628 nm.

## GST-D4H*-mCherry purification

pGEX-KG-D4H*-mCherry plasmid (Addgene #134604) was used to purify D4H*-mCherry as previously described (Lim et al., 2019). Briefly, a plasmid colony was grown in 5 mL LB broth overnight at 37 °C. Culture was scaled up into a 1 L flask and grown for a further 4 hr, and protein production induced with 0.5 mM IPTG for 4 hr. Bacterial culture was pelleted at 2500 rpm for 30 min and resuspended in lysis buffer (0.1 M NaCl, 20 mM Tris-HCl, pH 8.0, 1 mM DTT, 1 X protease inhibitor cocktail) and 0.35 mg/mL lysozyme added for 30 min on ice. Samples were sonicated at 50% Amp for 10 s and 10 s break for a total of 4 min. 1% Triton X-100 was added, and samples rocked at 4 °C for

30 min before centrifugation and incubation of supernatants with glutathione beads (GE17-0756-01, Sigma Aldrich) for 2 hr rocking at room temperature to capture the GST-D4H*-mCherry protein. D4H*-mCherry protein was eluted in elution buffer (25 mM L-glutathione, 50 mM Tris-HCl pH 8.8, 200 mM NaCl), filtered through a 30 kDa centrifugal filter unit (UFC203024, Thermo Fisher Scientific) and stored at –80 °C.

### D4H*-mCherry cell labelling and imaging

INS-1 832/3 SNAP/FLAG-hGLP-1R WT or V229A cells were seeded onto 13 mm coverslips and labelled with SNAP-Surface 488 probe in full media for 20 min before stimulation with 100 nM exendin-4 for 2 min at 37 °C. Cells were washed with PBS and fixed with 4% paraformaldehyde (PFA) for 10 min at 4 °C, permeabilised for 15 s in a liquid nitrogen bath, blocked in 1% BSA/PBS for 1 hr at room temperature, incubated with D4H*-mCherry (1:25 in 1% BSA / PBS) for 2 hr, washed with PBS and post-fixed in 4% PFA for 10 min at room temperature. Coverslips were washed, mounted onto microscope slides using ProLong Diamond Antifade Mountant with DAPI (P36966, Thermo Fisher Scientific), and imaged with a 63 X oil objective Leica Stellaris 8 inverted confocal microscope from the FILM facility at Imperial College London. Raw images were analysed in Fiji using the Coloc 2 plugin to determine SNAP/FLAG-hGLP-1R co-localisation with D4H*-mCherry.

### Raster image correlation spectroscopy (RICS) image capture and analysis

INS-1 832/3 SNAP/FLAG-hGLP-1R WT or V229A cells were seeded onto glass bottom MatTek dishes and labelled with SNAP-Surface 647 (S9136S, New England Biolabs) for 15 min at 37 °C in full media, washed, and imaged in RMPI-1640 media without phenol red using a 100 X oil objective on a Leica Stellaris 8 STED FALCON microscope from the FILM Facility at Imperial College London using confocal settings as previously described (*Bernabé-Rubio et al., 2021a*). Images were analysed as previously described (*Garcia and Bernardino de la Serna, 2018Pickford et al., 2020*). Briefly, cells were imaged at the basal plasma membrane under vehicle conditions and following stimulation with 100 nM exendin-4 with a format size of 256 × 256 pixels and 80 nm pixel size for 200 consecutive frames. RICS analysis was carried out to determine the diffusion coefficient of WT *vs* V229A GLP-1R in vehicle and stimulated conditions using the SimFCS 4 Software (Global Software, G-SOFT Inc). Three different regions of interest of size 32 × 32 pixels for each image and a moving average of 10 was applied to avoid any artefacts due to cellular motion or very slow-moving particles. Average intensity, intensity plots, 2D autocorrelation maps, and 3D autocorrelation fits were generated for each condition and cell line investigated (*Garcia and Bernardino de la Serna, 2018*; *Bernabé-Rubio et al., 2021b*).

### GLP-1R total internal reflection fluorescence (TIRF)-single particle tracking (SPT)

INS-1 832/3 GLP-1R KO cells were transiently transfected with WT or V229A human GLP-1R-monomeric EGFP fusion constructs, generated *in house* by mutating EGFP Ala207 to Lys in hGLP-1R-EGFP [kind gift from Dr Alessandro Bisello (*Syme et al., 2006*)] using the following primers: Forward (5' to 3'): CCTGAGCACCCAGTCCAAGCTGAGCAAAGACCCCA and Reverse (5' to 3'): TGGGGTCTTTGCTCAGCTTGGACTGGGTGCTCAGG, and WT hGLP-1R-mEGFP to V229A with the primers from *Table 2*. Transfected cells were seeded onto glass bottom MatTek dishes, imaged live at 37 °C in RPMI-1640 without phenol red under vehicle conditions or following stimulation with 1 µM exendin-4 and images analysed as previously described (*Salavessa and Sauvonnet, 2021*) with some changes. Cells were imaged in a Nikon spinning disk field scanning confocal microscope using a 100 X oil objective under TIRF illumination with the following microscope settings using the MetaMorph software: 488 nm excitation laser, exposure time of 150 ms, 10% laser power at a rate of 1 frame per second for 150 frames. Images were loaded onto the Icy software; an ROI was drawn around each cell and Spot Detector was used to determine the spots in each image. A scale of 2 with a sensitivity of 70 was used and kept constant for each experiment. Spot Tracking followed by Track Manager were used to determine the speed and displacement for each particle detected in the plasma membrane of cells.

## Raster image cross-correlation spectroscopy (RICCS) with number and brightness (N&B) image acquisition and data processing

INS-1 832/3 SNAP/FLAG-hGLP-1R WT or V229A cells were seeded onto μ-Slide 8-well glass bottom microscope plates (Ibidi) and labelled with SNAP-Surface Alexa Fluor 647 for 10 min followed by 40 μM Laurdan (6-dodecanoyl-2-(dimethylamino) naphthalene, D250, Thermo Fisher Scientific), an environmentally sensitive plasma membrane lipid dye (*Brewer et al., 2010*), for 5 min prior to imaging. Cells were washed with PBS and imaged in RMPI 1640 medium without phenol red using the confocal settings of a Leica Stellaris 8 STED FALCON in an 86 X objective and a 10% power 405 nm excitation laser and emission ranges of 420–460 nm (blue emission, IB) and 470–510 nm (red emission, IR) for Laurdan, plus a 5% power 647 nm excitation laser for SNAP-Surface Alexa Fluor 647. The general polarisation (GP) formula (GP = IB-IR/IB +IR) was used to retrieve the relative lateral packing order of lipids at the plasma membrane: values of GP vary from 1 to −1, with higher values reflecting lower fluidity (or higher lateral lipid order), whereas lower values indicate increased fluidity. Two hundred consecutive frames were captured for 2.45 min at a pixel size of 80 nm and a format size of 256 × 256 pixels. Cells were imaged under vehicle conditions and after stimulation with 100 nM exendin-4. Further analysis was carried out using the SimFCS 4 Software to determine the RICCS cross-correlation between Laurdan GP values and SNAP/FLAG-hGLP-1R. Specifically, Ch2-Ch1 (B1-B2 map) RICS analysis of brightness (intensity of fluorescence signal) in one channel cross-correlates with brightness in the second channel, allowing for the calculation of cross-correlation. N&B analysis was performed from the SNAP signal to determine the oligomerisation of SNAP/FLAG-hGLP-1R after exendin-4 stimulation by assessing the apparent brightness and number of pixels involved using SimFCS 4 software. Three different regions of interest of size 64 × 64 pixels per image and a moving average of 10 was applied to avoid any artefacts due to cellular motion or very slow-moving particles.

## GLP-1R time-resolved FRET (TR-FRET) conformational assay

INS-1 832/3 SNAP/FLAG-hGLP-1R WT or V229A cells were labelled with 40 nM SNAP-Lumi4-Tb (SSNPTBC, Cisbio), a lanthanide fluorophore probe for TR-FRET, for 1 hr at 37 °C in full media. Cells were washed with HBSS and resuspended in 60 nM NR12A, a solvatochromic photostable plasma membrane lipid probe (*Klymchenko, 2023*) in HBSS, incubated for 5 min, and seeded onto a white opaque 96-well half-area plate. TR-FRET was carried out as described (*Lucey et al., 2021*). Briefly, a baseline reading was taken for 5 min using a Flexstation 3 plate reader with excitation wavelength of 335 nm, and emissions of 490 nm and 590 nm. The cells were then stimulated with 100 nM exendin-4, or vehicle, and the plate read for a further 20 min. Receptor conformational shift was calculated as the 590/490 ratio of both wavelength emission signals after normalising to basal signal under vehicle conditions. AUC was calculated and used for statistical analysis.

## Lipid raft recruitment assay

INS-1 832/3 SNAP/FLAG-hGLP-1R WT or V229A cells were seeded onto 6 cm dishes and stimulated with 100 nM exendin-4 or vehicle for 2 min at 37 °C, placed on ice, and washed with ice cold PBS before being osmotically lysed with 20 mM Tris-HCl (pH 7.0) supplemented with protease and phosphatase inhibitor cocktails. The cell suspension was homogenized by passing through a 21-gauge needle and ultracentrifuged for 1 hr at 41,000 rpm at 4 °C. The supernatant was discarded, and the pellet resuspended in cold PBS +1% Triton X-100 plus protease and phosphatase inhibitor cocktails, transferred to an Eppendorf tube, and allowed to rotate for 30 min at 4 °C. The suspension was then ultracentrifuged for another hour at 41,000 rpm at 4 °C to separate detergent-resistant (DRM) from detergent-soluble (DSM) membrane fractions: the 'disordered' DSM fraction in the supernatant was retained for analysis and the 'ordered' DRM (lipid raft) fraction was resuspended in 1% SDS plus protease and phosphatase inhibitor cocktails, sonicated, centrifuged for 10 min at 13,500 rpm at 4 °C, and resolved via SDS-PAGE and Western blotting as in *Buenaventura et al., 2019*. Blotted membranes were analysed using Fiji and SNAP levels normalised to flotillin (detected post-stripping with anti-flotillin-1 antibody, sc-74566, Santa Cruz Biotechnology, RRID:AB_2106563), used as DRM loading control.

## GLP-1R co-localisation with clathrin-coated pits (CCPs) with using TIRF microscopy

INS-1 832/3 SNAP/FLAG-hGLP-1R WT or V229A cells were transfected with clathrin light chain (CLC)-GFP and plated onto poly-L-lysine-coated glass coverslips. Prior to imaging, cells were labelled at 37 °C with 5 µM SNAP-Surface 549 probe for 30 min in full media and washed. Microscopy was performed on a custom-built TIRF microscope based on a Zeiss Axiovert 135 with DPSS lasers at 405, 491, and 561 nm, 100 X/1.45 NA objective. Colour channels were split using a Dualview (Optical Insights) and projected side by side onto an EMCCD camera (QuantEM:512SC, Photometrics). Still images were acquired by sequential excitation with the 473 and 561 nm lasers. During the experiments, cells were incubated in an imaging solution containing 138 mM NaCl, 5.6 mM KCl, 2.6 mM $CaCl_2$, 1.2 mM $MgCl_2$, 3 mM glucose, and 5 mM HEPES, pH 7.4 at ~35 °C. Exendin-4 (100 nM) was added for at least 10 min to trigger receptor endocytosis. Calibration and alignment were performed using 100 nm polystyrene beads (Molecular Probes). To quantify SNAP/FLAG-hGLP-1R association with clathrin, circular ROIs surrounding individual well-defined puncta of fluorescence (5–50 per cell) were selected on CLC-GFP images and transferred to the corresponding SNAP ones using a Fiji macro and average fluorescence per pixel was measured in a central circle (c) of diameter 5 pixels (0.55 µm) at each puncta location, in a surrounding annulus (a) with an outer diameter of 7 pixels (0.77 µm), and in an area not containing any cell as background (bg). The specific cluster-associated fluorescence $\Delta F$ was calculated by subtracting the annulus value from that of the circle ($\Delta F = c$ a). Local fluorescence from CLC-GFP molecules not associated with the cluster site was calculated by subtracting the background from the annulus value ($S = a$ bg) and used to normalize $\Delta F$ to the expression level of the tested protein (*Barg et al., 2010*). The reported parameter $\Delta F/S$ can be interpreted as the GLP-1R affinity for clathrin structures.

## GLP-1R trafficking by high content microscopy

### SNAP/FLAG-hGLP-1R internalisation

INS-1 832/3 SNAP/FLAG-hGLP-1R WT or V229A cells were seeded onto poly-L-lysine-coated black, clear bottom 96-well plates. Cells were imaged and analysed as previously described (*Fang et al., 2020*). Briefly, cells were labelled with 1 µM BG-S-S-649, a cleavable SNAP-Surface probe, in full media for 30 min at 37 °C, washed with PBS, and stimulated with vehicle or 100 nM exendin-4 in reverse time order for 30, 15, 10 and 5 min at 37 °C in full media. Cells were washed with ice cold HBSS, and the following steps performed at 4 °C. Surface receptor probe was cleaved by incubation in 100 mM 2-mercaptoethane-sulfonic acid sodium salt (Mesna) in 1 X TAE buffer, pH 8.6 or incubated 1 X TAE buffer alone for 5 min and then washed and imaged in HBSS using Micro-Manager software in a Cairn Research widefield microscope with Nikon Ti2, CoolLED light source and 20 X phase contrast objective for both epifluorescence and transmitted phase contrast with 9 ROI acquired per well. Surface receptor fluorescence was quantified after flat-field correction using BaSiC and cell-containing regions were segmented using Phantast with the following script:

```
run('32-bit');
run('PHANTAST,' 'sigma = 4 epsilon = 0.08 new');
```

Mean intensity for the cell-containing region was determined using a custom macro by Steven Rothery, Imperial College London FILM Facility (https://www.imperial.ac.uk/medicine/facility-for-imaging-by-light-microscopy/software/fiji/). SNAP/FLAG-hGLP-1R internalisation was calculated by first subtracting non-specific signal of wells treated with Mesna without agonist from all the other wells, and then normalising to signal of cells with no Mesna exposure (*Jones et al., 2021*).

### SNAP/FLAG-hGLP-1R plasma membrane recycling

INS-1 832/3 SNAP/FLAG-hGLP-1R WT or V229A cells were seeded onto poly-L-lysine-coated black, clear bottom 96-well plates in full media, stimulated with vehicle or 100 nM exendin-4 for 1 hr at 37 °C to enable receptor internalisation, washed with PBS and incubated in 100 nM tetramethylrhodamine (TMR)-tagged exendin-4 (Ex-4-TMR) in full media for 1 hr or 3 hr at 37 °C to label any receptors that recycle back to the plasma membrane after agonist-mediated internalisation. Cells were washed, imaged in HBSS, and processed as above with Ex-4-TMR fluorescence signal corresponding to plasma

membrane recycled receptor along with low signals of residual surface receptor (*Pickford et al., 2020*). Background/non-specific fluorescence from wells not treated with Ex-4-TMR was subtracted from all wells followed by normalising to signal from wells not treated with exendin-4 (vehicle conditions).

## SNAP/FLAG-hGLP-1R plasma membrane degradation

INS-1 832/3 SNAP/FLAG-hGLP-1R WT or V229A cells were seeded onto poly-L-lysine-coated black, clear bottom 96-well plates in full media and treated as previously described (*Manchanda et al., 2023*). Briefly, cells were washed with PBS and changed to media with no serum +50 µg/mL cycloheximide to inhibit protein synthesis for 1 hr, then stimulated with vehicle or 100 nM exendin-4 in reverse time order for 8, 6, 4, 2, and 1 hr. Media was changed in the last 30 min to full media with vehicle or agonist and 1 µM BG-OG, a permeable SNAP-tag probe to label total residual SNAP/FLAG-hGLP-1R. Cells were washed and imaged in HBSS as above. SNAP/FLAG-hGLP-1R degradation is quantified as the inverse of total residual receptor signal normalised to non-stimulated wells after subtracting non-labelled background signal.

## NanoBiT complementation assays

GLP-1R-SmBiT WT was previously cloned *in house* as described in *Manchanda et al., 2023*. GLP-1R-SmBiT V229A was generated from the WT template by site-directed mutagenesis using the following primers: Forward (5' to 3'): TCTTTCCTGTCGACTCGCGTTTCTGCTTATGCAGT, Reverse (5' to 3'): ACTGCATAAGCAGAAACGCGAGTCGACAGGAAAG. LgBiT-mini-$G_s$ and -mini-$G_q$ were a gift from Prof. Nevin Lambert, Medical College of Georgia, USA, and LgBiT-β-arrestin 2 was purchased from Promega (plasmid no. CS1603B118). For recruitment assays, INS-1 832/3 GLP-1R KO cells were seeded onto 6-well plates and transiently transfected with a combination of 1 µg GLP-1R-SmBiT WT or V229A and either 1 µg of LgBiT-mini-$G_s$, -mini-$G_q$ or -β-arrestin 2.

Nanobody37 (Nb37) bystander NanoBiT assays (*Figure 7—figure supplement 1*) were carried out as previously described (*Manchanda et al., 2023*). The constructs required to determine plasma membrane (KRAS CAAX motif), or endosomal (Endofin FYVE domain) GLP-1R activity were a gift from Prof. Asuka Inoue, Tohoku University, Japan. INS-1 832/3 GLP-1R KO cells were seeded onto 12-well plates and transfected as detailed in *Table 3*.

Briefly, 24 hr after transfection, cells were incubated in 3 mM glucose for a minimum of 2 hr, detached and resuspended in NanoGlo Live cell substrate (N2012, Promega), 1:20 dilution in HBSS, and seeded onto an opaque white 96-well half-area plate. Using a Flexstation 3 plate reader, a baseline luminescence reading was acquired for 8 min at 37°C, followed by stimulation with vehicle or 100 nM exendin-4 for Nb37 bystander assay, or with serial dilutions of exendin-4 starting from 1 µM, 100 nM, 10 nM to 1 nM for 30 min for the mini-$G_s$/mini-$G_q$/β-arrestin 2 recruitment assays. Results were normalised to individual basal signal followed by average vehicle responses. Non-linear fit curves of the AUCs were used to generate dose responses using GraphPad Prism 10.

**Table 3.** Transfected plasmid amounts for Nb37 bystander NanoBiT assays.

| LgBiT-KRAS CAAX motif (SSSGGGKKKKKKKSKTKCVIM) | | Endofin FYVE domain-LgBiT (amino acid region Gln739-Lys806) | |
|---|---|---|---|
| Plasmid | Concentration | Plasmid | Concentration |
| SNAP/FLAG-hGLP-1R WT or V229A | 0.1 µg | SNAP/FLAG-hGLP-1R WT or V229A | 0.1 µg |
| Nb37-SmBiT | 0.05 µg | Nb37-SmBiT | 0.25 µg |
| LgBiT-CAAX | 0.05 µg | Endofin-LgBiT | 0.25 µg |
| G$\alpha_s$ (human, short isoform) | 0.25 µg | G$\alpha_s$ (human, short isoform) | 0.25 µg |
| Gβ1 (human) | 0.25 µg | Gβ1 (human) | 0.25 µg |
| Gγ1 (human) | 0.25 µg | Gγ1 (human) | 0.25 µg |
| RIC8B (human, isoform 2) | 0.05 µg | RIC8B (human, isoform 2) | 0.05 µg |
| pcDNA 3.1 | 0.40 µg | pcDNA 3.1 | - |

## Statistical analyses

All data analyses and graph generation were performed using GraphPad Prism 10 and presented as mean ± standard error of the mean (SEM); *n* numbers represent biological replicates. Statistical analyses were performed with the tests indicated in the figure legend. Data was deemed significant at p≤0.05.

## Acknowledgements

The authors thank the FILM Facility, Imperial College London, for technical support on microscopy experiments and microscopy data analysis, Prof Mark Sansom and Dr Wanling Song, Dept Biochemistry, University of Oxford, for assistance with the use of PyLipID for the GLP-1R-cholesterol binding site analysis, and Dr Aida Martinez-Sanchez, Imperial College London, for providing access to the Home Office PPL animal license. The A.T. group is funded by grants from Diabetes UK (19/0006094), the MRC (MR/X021467/1), and the Wellcome Trust (301619/Z/23/Z), the latter in collaboration with S.L.R, B.J. and J.B.S; A.I.O. is supported by a PhD Scholarship from the Commonwealth. The S.L.R. group is supported by MRC grant MR/T017961/1. The S.B. group is supported by the Swedish Research Council, Diabetes Wellness Sweden, Novo Nordisk Foundation, and the Swedish Diabetes Foundation. The J.B.S. lab acknowledges funding from BBSRC (BB/V019791/1) and MRC (MR/W024985/1). B.J. is supported by the MRC Clinician Scientist Fellowship MR/Y00132X/1, and by project grants from the MRC and Diabetes UK. The Section of Endocrinology at Imperial College London is funded by grants from the MRC, NIHR and is supported by the NIHR Biomedical Research Centre Funding Scheme and the NIHR/Imperial Clinical Research Facility. The views expressed are those of the author(s) and not necessarily those of the NHS, the NIHR, or the Department of Health.

## Additional information

### Funding

| Funder | Grant reference number | Author |
| --- | --- | --- |
| Wellcome Trust | 301619/Z/23/Z | Alejandra Tomas<br>Ben Jones<br>Jorge Bernardino de la Serna<br>Sarah L Rouse |
| Medical Research Council | MR/X021467/1 | Alejandra Tomas<br>Ben Jones |
| Diabetes UK | 19/0006094 | Alejandra Tomas |
| Commonwealth | PhD Scholarship | Affiong Ika Oqua |
| Medical Research Council | MR/T017961/1 | Sarah L Rouse |
| Swedish Research Council | | Sebastian Barg |
| Diabetes Wellness Sweden | | Sebastian Barg |
| Novo Nordisk Foundation | | Sebastian Barg |
| Swedish Diabetes Foundation | | Sebastian Barg |
| BBSRC | BB/V019791/1 | Jorge Bernardino de la Serna |
| Medical Research Council | MR/W024985/1 | Jorge Bernardino de la Serna |

The funders had no role in study design, data collection and interpretation, or the decision to submit the work for publication. For the purpose of Open Access, the authors have applied a CC BY public copyright license to any Author Accepted Manuscript version arising from this submission.

### Author contributions
Affiong Ika Oqua, Formal analysis, Investigation, Visualization, Writing – original draft; Kin Chao, Liliane El Eid, Investigation, Visualization; Lisa Casteller, Billy P Baxter, Alba Miguéns-Gómez, Investigation; Sebastian Barg, Ben Jones, Supervision, Visualization; Jorge Bernardino de la Serna, Formal analysis, Supervision, Visualization; Sarah L Rouse, Data curation, Formal analysis, Supervision, Validation, Investigation, Visualization, Methodology; Alejandra Tomas, Conceptualization, Resources, Data curation, Formal analysis, Supervision, Funding acquisition, Validation, Investigation, Visualization, Methodology, Writing – original draft, Project administration, Writing - review and editing

### Author ORCIDs
Sebastian Barg (iD) https://orcid.org/0000-0003-4661-5724
Jorge Bernardino de la Serna (iD) https://orcid.org/0000-0002-1396-3338
Sarah L Rouse (iD) http://orcid.org/0000-0002-7115-1565
Alejandra Tomas (iD) https://orcid.org/0000-0002-2290-8453

### Ethics
In vivo studies were carried out under the approval of the UK Home Office Animals (Scientific Procedures) Act 1986 Project Licence PP7151519 (PPL Holder: Dr A. Martinez-Sanchez, Imperial College London) at the Central Biological Services unit of Imperial College London.

Reviewer #1 (Public review): https://doi.org/10.7554/eLife.101011.3.sa1
Reviewer #2 (Public review): https://doi.org/10.7554/eLife.101011.3.sa2
Author response https://doi.org/10.7554/eLife.101011.3.sa3

---

# Additional files

### Supplementary files
MDAR checklist

Supplementary file 1. Active GLP-1R - cholesterol interactions.

Supplementary file 2. Inactive GLP-1R - cholesterol interactions.

Supplementary file 3. Insulin secretion raw data.

### Data availability
All data generated or analysed during this study are included in the manuscript and supporting files.

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
