## [Editor Report · eLife Assessment]

The study presents a **valuable** finding on the role of cholesterol-binding sites on GLP-1 receptors although the clinical ramifications are unclear and not eminent at this point. Based on the detailed and persuasive responses provided by authors to the concerns raised by reviewers, the revised manuscript is improved substantially and is **convincing** enough in its scientific merit. The study is a good addition to the scientific community working on receptor biology and drug development for GLP-1 R.

---

## [Referee Report · Reviewer #1 (Public review)]

Summary:

The authors demonstrate impairments induced by a high cholesterol diet on GLP-1R dependent glucoregulation in vivo as well as an improvement after reduction in cholesterol synthesis with simvastatin in pancreatic islets. They also map sites of cholesterol high occupancy and residence time on active versus inactive GLP-1Rs using coarse-grained molecular dynamics (cgMD) simulations, and screened for key residues selected from these sites and performed detailed analyses of the effects of mutating one of these residues, Val229, to alanine on GLP-1R interactions with cholesterol, plasma membrane behaviour, clustering, trafficking and signalling in pancreatic beta cells and primary islets, and describe an improved insulin secretion profile for the V229A mutant receptor.

These are extensive and very impressive studies indeed. I am impressed with the tireless effort exerted to understand the details of molecular mechanisms involved in the effects of cholesterol for GLP-1 activation of its receptor. In general, the study is convincing, the manuscript well written and the data well presented. Some of the changes are small and insignificant which makes one wonder how important the observations are. For instance, in Figure 2E (which is difficult to interpret anyway because the data are presented in per cent, conveniently hiding the absolute results) does not show a significant result of the cyclodextrin except for insignificant increases in basal secretion. That is not identical to impairment of GLP-1 receptor signaling!

To me the most important experiment of them all is the simvastatin experiment, but the results rest on very few numbers and there is a large variation. Apparently, in a previous study using more extensive reduction in cholesterol the opposite response was detected casting doubt on the significance of the current observation. I agree with the authors that the use of cyclodextrin may have been associated with other changes in plasma membrane structure than cholesterol depletion at the GLP-1 receptor. The entire discussion regarding the importance of cholesterol would benefit tremendously from studies of GLP-1 induced insulin secretion in people with different cholesterol levels before and after treatment with cholesterol-lowering agents. I suspect that such a study would not reveal major differences.

Comments on revisions: The authors have responded well to my criticism.

---

## [Referee Report · Reviewer #2 (Public review)]

Summary:

In this manuscript the authors were providing a proof of concept that they can identify and mutate a cholesterol-binding site of a high-interest class B receptor, the GLP-1R, and functionally characterize the impact of this mutation on receptor behavior in the membrane and downstream signaling with the intent that similar methods can be useful to optimize small molecules that as ligands or allosteric modulators of GLP-1R can improve the therapeutic tools targeting this signaling system.

Strengths:

The majority of results on receptor behavior are elucidated in INS-1 cells expressing the wt or mutant GLP-1R, with one experiment translating the findings to primary mouse beta-cells. I think this paper lays a very strong foundation to characterize this mutation and does a good job discussing how complex cholesterol-receptor interactions can be (ie lower cholesterol binding to V229A GLP-1R, yet increased segregation to lipid rafts). Table 1 and Figure 9 are very beneficial to summarize the findings. The lower interaction with cholesterol and lower membrane diffusion in V229A GLP-1R resembles the reduced diffusion of wt GLP-1R with simv-induced cholesterol reductions, by presumably decreasing the cholesterol available to interact with wt GLP-1R. The effects of this mutation are not due to differences in Ex-4:recepotor affinity. I think this paper will be of interest to many physiologists who may not be familiar with many of the techniques used in this paper and the authors largely do a good job explaining the goals of using each method in the results section. While not necessary for this paper, a comparison of islet cholesterol content after this cholesterol diet vs the more typical 60% HFD used in obesity research would be beneficial for GLP-1 physiology research broadly to take these findings into consideration with model choice.

Weaknesses:

There are no obvious weaknesses in this manuscript and overall, I believe the authors achieved their aims and have demonstrated the importance of cholesterol interactions on GLP-1R functioning in beta-cells.

Certainly many follow-up experiments are possible from these initial findings and of primary interest is how this mutation affects insulin homeostasis in vivo under different physiological conditions. One of the biggest pathologies in insulin homeostasis in obesity/t2d is an elevation of baseline insulin release (as modeled in Fig 1E) that renders the fold-change in glucose stimulated insulin levels lower and physiologically less effective. Future work by the authors may determine the effects of the GLP-1R V229A mutation on insulin secretion responses under diet-induced metabolic stress conditions. Furthermore, the authors may additionally investigate if V229A would have the same impact in a different cell type, especially in neurons, with implications in the regulation of satiation, gut motility, and especially nausea, which are of high translational interest.

The comparison is drawn in the discussion between this mutation and ex4-phe1 to have biased agonism towards Gs over beta-arrestin signaling. Ex4-phe1 lowered pica behavior (a proxy for nausea) in the authors previously co-authored paper on ex4-phe1 (PMID 29686402) and drawing a parallel for this mutation or modification of cholesterol binding to potentially mitigate nausea is a novel direction.

---

## [Author Response]

The following is the authors’ response to the original reviews

**Public Reviews:**

**Reviewer #1 (Public review):**
Summary:The authors demonstrate impairments induced by a high cholesterol diet on GLP-1R dependent glucoregulation in vivo as well as an improvement after reduction in cholesterol synthesis with simvastatin in pancreatic islets. They also map sites of cholesterol high occupancy and residence time on active versus inactive GLP-1Rs using coarse-grained molecular dynamics (cgMD) simulations and screened for key residues selected from these sites and performed detailed analyses of the effects of mutating one of these residues, Val229, to alanine on GLP-1R interactions with cholesterol, plasma membrane behaviour, clustering, trafficking and signalling in pancreatic beta cells and primary islets, and describe an improved insulin secretion profile for the V229A mutant receptor.These are extensive and very impressive studies indeed. I am impressed with the tireless effort exerted to understand the details of molecular mechanisms involved in the effects of cholesterol for GLP-1 activation of its receptor. In general, the study is convincing, the manuscript well written and the data well presented.Some of the changes are small and insignificant which makes one wonder how important the observations are. For instance, in figure 2 E (which is difficult to interpret anyway because the data are presented in percent, conveniently hiding the absolute results) does not show a significant result of the cyclodextrin except for insignificant increases in basal secretion. That is not identical to impairment of GLP-1 receptor signaling!

We assume that the reviewer refers to Figure 1E, where we show the percentage of insulin secretion in response to 11 mM glucose +/- exendin-4 stimulation in mouse islets pretreated with vehicle or MβCD loaded with 20 mM cholesterol. While we concur with the reviewer that the effect in this case is triggered by increased basal insulin secretion at 11 mM glucose, exendin-4 appears to no longer compensate for this increase by proportionally amplifying insulin responses in cholesterol-loaded islets, leading to a significantly decreased exendin-4induced insulin secretion fold increase under these circumstances, as shown in Figure 1F. We interpret these results as a defect in the GLP-1R capacity to amplify insulin secretion beyond the basal level to the same extent as in vehicle conditions. An alternative explanation is that there is a maximum level of insulin secretion in our cells, and 11 mM glucose + exendin-4 stimulation gets close to that value. With the increasing effect of cholesterol-loaded MβCD on basal secretion at 11 mM glucose, exendin-4 stimulation would then appear to work less well.

We have performed a simple experiment to investigate this possibility: insulin secretion following stimulation with a secretagogue cocktail (20 mM glucose, 30 mM KCl, 10 µM FSK and 100 µM IBMX) in islets +/- MβCD/cholesterol loading to determine if maximal stimulation had been reached or not in our original experiment. This experiment, now included in Figure 1 - Figure Supplement 1C, demonstrates that insulin secretion can increase up to ~4% (from ~2%) in our islets, supporting our initial conclusion. We have also included absolute insulin concentrations as well as percentages of secretion for all the experiments included in the study in the new Supplementary File 3 to improve the completeness of the report.

To me the most important experiment of them all is the simvastatin experiment, but the results rest on very few numbers and there is a large variation. Apparently, in a previous study using more extensive reduction in cholesterol the opposite response was detected casting doubt on the significance of the current observation. I agree with the authors that the use of cyclodextrin may have been associated with other changes in plasma membrane structure than cholesterol depletion at the GLP-1 receptor.

We agree with the reviewer that the insulin secretion results in vehicle *versus* LPDS/simvastatin treated mouse islets (Figure 1H, I) are relatively variable. We have therefore performed 2 extra biological repeats of this experiment (for a total n of 7). Results now show a significant increase in exendin-4-stimulated secretion with no change in basal secretion in islets pre-incubated with LPDS/simvastatin.

The entire discussion regarding the importance of cholesterol would benefit tremendously from studies of GLP-1 induced insulin secretion in people with different cholesterol levels before and after treatment with cholesterol-lowering agents. I suspect that such a study would not reveal major differences.

We agree with the reviewer that such study would be highly relevant. While this falls outside the scope of the present paper, we encourage other researchers with access to clinical data on GLP-1R agonist responses in individuals taking cholesterol lowering agents to share their results with the scientific community. We have highlighted this point in the paper discussion to emphasise the importance of more research in this area.

**Reviewer #2 (Public review):**
Summary:In this manuscript the authors provided a proof of concept that they can identify and mutate a cholesterol-binding site of a high-interest class B receptor, the GLP-1R, and functionally characterize the impact of this mutation on receptor behavior in the membrane and downstream signaling with the intent that similar methods can be useful to optimize small molecules that as ligands or allosteric modulators of GLP-1R can improve the therapeutic tools targeting this signaling system.Strengths:The majority of results on receptor behavior are elucidated in INS-1 cells expressing the wt or mutant GLP-1R, with one experiment translating the findings to primary mouse beta-cells. I think this paper lays a very strong foundation to characterize this mutation and does a good job discussing how complex cholesterol-receptor interactions can be (ie lower cholesterol binding to V229A GLP-1R, yet increased segregation to lipid rafts). Table 1 and Figure 9 are very beneficial to summarize the findings. The lower interaction with cholesterol and lower membrane diffusion in V229A GLP-1R resembles the reduced diffusion of wt GLP-1R with simv-induced cholesterol reductions, although by presumably decreasing the cholesterol available to interact with wt GLP-1R. This could be interesting to see if lowering cholesterol alters other behaviors of wt GLP-1R that look similar to V229A GLP-1R. I further wonder if the authors expect that increased cholesterol content of islets (with loading of MβCD saturated with cholesterol or high-cholesterol diets) would elevate baseline GLP-1R membrane diffusion, and if a more broad relationship can be drawn between GLP-1R membrane movement and downstream signaling.

Membrane diffusion experiments are difficult to perform in intact islets as our method requires cell monolayers for RICS analysis. We however agree that it is of interest to investigate if cholesterol loading affects GLP-1R diffusion. To this end, we have performed further RICS analysis in INS-1 832/3 SNAP/FLAG-hGLP-1R cells pretreated with vehicle or MβCD loaded with 20 mM cholesterol (new Figure 1 - Figure Supplement 1D and 1E). Interestingly, results show significantly increased plasma membrane diffusion of exendin-4-stimulated receptors, with no change in basal diffusion, following MβCD/cholesterol loading. This behaviour differs from that of the V229A mutant receptor which shows reduced diffusion under basal conditions, a pattern that mimics that of the WT receptor under low cholesterol conditions (by pre-treatment with LPDS/simvastatin).

Weaknesses:I think there are no obvious weaknesses in this manuscript and overall, I believe the authors achieved their aims and have demonstrated the importance of cholesterol interactions on GLP-1R functioning in beta-cells. I think this paper will be of interest to many physiologists who may not be familiar with many of the techniques used in this paper and the authors largely do a good job explaining the goals of using each method in the results section. The intent of some methods, for example the Laurdan probe studies, are better expanded in the discussion.I found it unclear what exactly was being measured to assess 'receptor activity' in Fig 7E and F.

We have expanded on the rationale behind the use of Laurdan to assess behaviours of lipid packed membrane nanodomains in the methods, results and discussion of the revised manuscript.

Figures 7E and F refer to bystander complementation assays measuring the recruitment of nanobody 37 (Nb37)-SmBiT, which binds to active Gas, to either the plasma membrane (labelled with KRAS CAAX motif-LgBiT), or to endosomes (labelled with Endofin FYVE domain-LgBiT) in response to GLP-1R stimulation with exendin-4. This assay therefore measures GLP-1R activation specifically at each of these two subcellular locations. We have included a schematic of this assay in the new Figure 7 - Figure Supplement 1 to clarify the aim of these experiments.

Certainly many follow-up experiments are possible from these initial findings and of primary interest is how this mutation affects insulin homeostasis in vivo under different physiological conditions. One of the biggest pathologies in insulin homeostasis in obesity/t2d is an elevation of baseline insulin release (as modeled in Fig 1E) that renders the fold-change in glucose stimulated insulin levels lower and physiologically less effective. No difference in primary mouse islet baseline insulin secretion was seen here but I wonder if this mutation would ameliorate diet-induced baseline hyperinsulinemia.

We concur with the reviewer that it would be interesting to determine the effects of the GLP1R V229A mutation on insulin secretion responses under diet-induced metabolic stress conditions. While performing in vivo experiments on glucoregulation in mice harbouring the V229A mutation falls outside the scope of the present study, we have included ex vivo insulin secretion experiments in islets from GLP-1R KO mice transduced with adenoviruses expressing SNAP/FLAG-hGLP-1R WT or V229A and subsequently treated with vehicle *versus* MβCD loaded with 20 mM cholesterol to replicate the conditions of Figure 1E in the new Figure 8 - Figure Supplement 1.

I would have liked to see the actual islet cholesterol content after 5wks high-cholesterol diet measured to correlate increased cholesterol load with diminished glucose-stimulated inulin. While not necessary for this paper, a comparison of islet cholesterol content after this cholesterol diet vs the more typical 60% HFD used in obesity research would be beneficial for GLP-1 physiology research broadly to take these findings into consideration with model choice.

We have included these data in Figure 1 - Figure Supplement 1A.

Another area to further investigate is does this mutation alter ex4 interaction/affinity/time of binding to GLP-1 or are all of the described findings due to changes in behavior and function of the receptor?

To answer this question, have performed binding affinity experiments, which show no differences, in INS-1 832/3 SNAP/FLAG-hGLP-1R WT *versus* V229A cells (new Figure 3 - Figure Supplement 1D).

Lastly, I wonder if V229A would have the same impact in a different cell type, especially in neurons? How similar are the cholesterol profiles of beta-cells and neurons? How this mutation (and future developed small molecules) may affect satiation, gut motility, and especially nausea, are of high translational interest. The comparison is drawn in the discussion between this mutation and ex4-phe1 to have biased agonism towards Gs over beta-arrestin signaling. Ex4-phe1 lowered pica behavior (a proxy for nausea) in the authors previously co-authored paper on ex4-phe1 (PMID 29686402) and I think drawing a parallel for this mutation or modification of cholesterol binding to potentially mitigate nausea is worth highlighting.

While experiments in neurons are outside the scope of the present study, we have added this worthy point to the discussion and hypothesise on possible effects of GLP-1R mutants with modified cholesterol interactions on central GLP-1R actions in the revised manuscript.

**Reviewer #1 (Recommendations for the authors):**
There are no line numbers

These have now been added.

Abstract: "Cholesterol is a plasma membrane enriched lipid" - sorry for being finicky, but shouldn't this read; "a lipid often enriched in plasma membranes"

We have modified the abstract to state that: “Cholesterol is a lipid enriched at the plasma membrane”.

p. 4 "Moreover, islets extracted from high cholesterol-fed mice". How do you "extract islets"?

We have exchanged the term “extracted” by “isolated”. Islet isolation is described in the paper methods section.

p. 4 The sentence "These effects were accompanied by decreased GLP-1R plasma membrane diffusion under vehicle conditions, measured by Raster Image Correlation Spectroscopy (RICS) in rat insulinoma INS-1 832/3 cells with endogenous GLP-1R deleted [INS-1 832/3 GLP-1R KO cells (27)] stably expressing SNAP/FLAG-tagged human GLP-1R (SNAP/FLAG-hGLP-1R), an effect that is normally triggered by agonist binding (28), as also observed here (Supplementary Figure 1C, D)" is a masterpiece of complexity. Perhaps breaking up would facilitate reading?

This paragraph has now been modified in the revised manuscript.

p. 5. I cannot evaluate the "coarse grain molecular dynamics" studies.
**Reviewer #2 (Recommendations for the authors):**
I view this as an excellent manuscript with very comprehensive work and clear translational relevance. I don't think any further experiments are needed for the scope outlined in this manuscript. The discussion is already long but a short postulation on how this may translate to GLP-1R-cholesterol interactions in other cell types, specifically neurons with the intent on manipulating satiation and nausea, could be worthwhile.

This has now been added.

The only thing for readability I would suggest is a sentence in the results mentioning why you're doing the Laurdan analysis, and what is the output for assessing 'receptor activity' in the membrane and endosomes.

Both points have now been added.